# Elucidating the path to *Plasmodium* prolyl-tRNA synthetase inhibitors that overcome halofuginone resistance

Mark A. Tye [1,2,3], N. Connor Payne [1,4,13], Catrine Johansson [5,6,13], Kritika Singh [1,7], Sofia A. Santos [1], Lola Fagbami [1,2,3,8], Akansha Pant [3], Kayla Sylvester [9], Madeline R. Luth [10], Sofia Marques [11], Malcolm Whitman [12], Maria M. Mota [11], Elizabeth A. Winzeler [10], Amanda K. Lukens [8], Emily R. Derbyshire [9], Udo Oppermann [5,6], Dyann F. Wirth [3,8] & Ralph Mazitschek [1,3,8] ✉

The development of next-generation antimalarials that are efficacious against the human liver and asexual blood stages is recognized as one of the world's most pressing public health challenges. In recent years, aminoacyl-tRNA synthetases, including prolyl-tRNA synthetase, have emerged as attractive targets for malaria chemotherapy. We describe the development of a single-step biochemical assay for *Plasmodium* and human prolyl-tRNA synthetases that overcomes critical limitations of existing technologies and enables quantitative inhibitor profiling with high sensitivity and flexibility. Supported by this assay platform and co-crystal structures of representative inhibitor-target complexes, we develop a set of high-affinity prolyl-tRNA synthetase inhibitors, including previously elusive aminoacyl-tRNA synthetase triple-site ligands that simultaneously engage all three substrate-binding pockets. Several compounds exhibit potent dual-stage activity against *Plasmodium* parasites and display good cellular host selectivity. Our data inform the inhibitor requirements to overcome existing resistance mechanisms and establish a path for rational development of prolyl-tRNA synthetase-targeted anti-malarial therapies.

Malaria is an infectious disease caused by *Plasmodium* parasites and ranks third among deadly infectious diseases, with over 200 million cases and more than 600,000 deaths per year[1]. The emergence and spread of resistance to all first-line antimalarials threatens our ability to treat and contain malaria. This problem is exacerbated by the limited number of targets exploited by current drugs, most of which are only relevant for the asexual blood stage (ABS), restricting their utility to the treatment of acute malaria. Therefore, new antimalarial therapies that exploit novel targets and pathways essential for multiple life-cycle stages are highly

[1]Center for Systems Biology, Massachusetts General Hospital, Boston, MA, USA. [2]Harvard Graduate School of Arts and Sciences, Cambridge, MA, USA. [3]Harvard T.H. Chan School of Public Health, Boston, MA, USA. [4]Department of Chemistry and Chemical Biology, Harvard University, Cambridge, MA, USA. [5]Botnar Research Centre, NIHR Oxford Biomedical Research Unit, University of Oxford, Oxford, UK. [6]Centre for Medicines Discovery, University of Oxford, Oxford, UK. [7]Department of Bioengineering, Northeastern University, Boston, MA, USA. [8]Broad Institute of MIT and Harvard, Cambridge, MA, USA. [9]Department of Chemistry, Duke University, Durham, NC, USA. [10]Department of Pediatrics, University of California, San Diego, La Jolla, CA, USA. [11]Instituto de Medicina Molecular, Faculdade de Medicina, Universidade de Lisboa, Lisbon, Portugal. [12]Department of Developmental Biology, Harvard School of Dental Medicine, Boston, MA, USA. [13]These authors contributed equally: N. Connor Payne, Catrine Johansson. ✉e-mail: ralph@broadinstitute.org

sought after for primary prophylaxis and transmission blocking, in addition to acute treatment[2,3].

Halofuginone (**1**) (Fig. 1a) is one of the most potent known antimalarials and a synthetic derivative of the natural product febrifuga, the curative ingredient of an ancient herbal remedy that has been used in Traditional Chinese Medicine for over 2000 years for the treatment of fevers and malaria[4,5]. However, the therapeutic utility of halofuginone and analogs as antimalarials has been stymied by poor tolerability, and the previously unknown mode of action in the host and parasite has impeded rational development of drugs with improved pharmacological properties[6].

Previously, we have identified the cytoplasmic prolyl-tRNA synthetase (ProRS) as the molecular target of halofuginone in *P. falciparum* (*Pf*cProRS, *PF3D7_1213800*). (Following the increasingly adapted convention, we will use the 3-letter amino acid code followed by RS for protein names of specific aminoacyl-tRNA synthetase (aaRS) isoforms)[7,8]. ProRS is a member of the aaRS enzyme family, which exist in all living cells and catalyze the transfer of amino acids to their cognate tRNAs[9]. However, recent research has also revealed secondary functions of specific aaRS isoforms and tRNAs beyond their canonical role in protein biosynthesis[10,11].

In addition, we have shown that halofuginone and derivatives are also active against liver stage parasites in vitro and in vivo, further validating *Pf*cProRS as an attractive target for antimalarial drug development[7,8]. In complementary efforts, investigating the mode of action of halofuginone in humans, where halofuginone has been studied as chemotherapeutic, antifibrotic, immunomodulatory agent and more recently as antiviral drug[12,13], we identified the ProRS activity of the bifunctional glutamyl-ProRS (*Hs*GluProRS) as the mechanistic target[14]. Crystallographic data of the co-complexes with human and *Plasmodium* ProRS revealed that halofuginone binds the A76-tRNA^Pro and proline-binding pockets of the active site (Fig. 1a), which are highly conserved between both homologs[15,16]. Despite the high homology

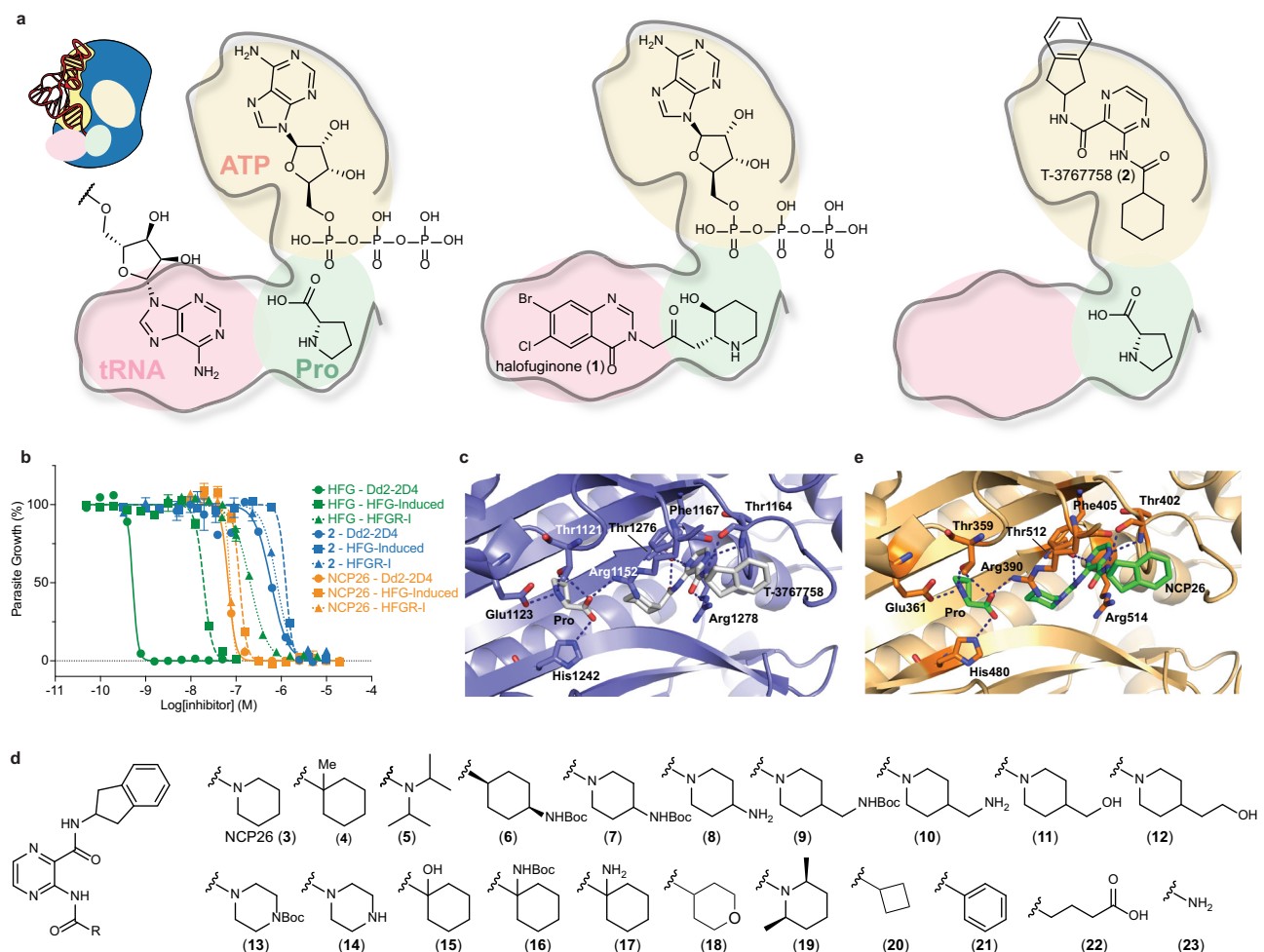

**Fig. 1 | ProRS inhibitor design and anti-*Plasmodium* activity. a** Schematic representation of the ProRS active site and binding mode of canonical substrates (proline, ATP, tRNA^Pro), halofuginone (**1**), and T-3767758 (**2**). The active site of ProRS constitutes three distinct substrate pockets, which bind the terminal adenosine (A76) residue of tRNA^Pro (red), proline (green), and ATP (yellow). Halofuginone binds in the tRNA^Pro and proline-binding pockets and requires the presence of ATP for tight binding (ATP-uncompetitive), while compound **2** targets the ATP-binding pocket and requires the presence of proline for tight binding (proline-uncompetitive). **b** In vitro activity of halofuginone (HFG, green), **2** (blue), and NCP26 (**3**, orange) against wild-type (Dd2-2D4, solid line), halofuginone-induced (HFG-induced, dashed line), and HFGR-I (dotted line) *P. falciparum* strains. Asexual blood stage (ABS) *P. falciparum* parasites were treated with test compounds at varying concentrations for 72 h, followed by quantification of parasite growth by SYBR Green staining. Compound **2** and NCP26 are refractory to halofuginone resistance. **c** Ribbon representation of *Hs*ProRS (blue ribbon/sticks) in complex with proline and compound **2** (silver sticks, PDB: 5VAD) reveals an energetically unfavorable axial confirmation of the cyclohexyl substituent. **d** Chemical structures of selected pyrazinamide analogs (see Supplementary Fig. 3 for additional ligands). **e** Co-crystal structure of proline and NCP26 (green sticks) bound to *Pf*cProRS (orange sticks, PDB: 6T7K) shows that the piperidine adopts an energetically more favorable confirmation, with no notable change in the interacting amino acid residues. Data in **b** are expressed as mean ± s.d. (*n* = 3 independent replicate wells) and are representative of ≥5 independent experiments.

between parasite and host enzymes (see Supplementary Fig. 1) and comparable biochemical potency, halofuginone is significantly more active against asexual blood-stage *P. falciparum* than mammalian cell lines[6,8,15].

More recently, when we investigated the evolution of halofuginone resistance, we discovered that within five generations (10 days), bulk cultures, termed halofuginone-induced parasites, mounted a 10–20-fold tolerance by upregulation (~20-fold) of intracellular proline, which is competitive with halofuginone[17]. This previously unrecognized mode of resistance could potentially also explain the failure of febrifugine and halofuginone to control recrudescence in vivo and their narrow therapeutic indices as antimalarials[6]. While these findings have validated *Pf*cProRS as a well-defined target for the rational development of next-generation antimalarials with multi-stage activity, the development of rapid resistance to halofuginone analogs has raised concerns.

Furthermore, a general problem that has plagued aaRS inhibitor discovery is the lack of robust, simple yet sensitive high-throughput assays that are suitable for supporting rational drug development[18–22]. Although several biochemical assay platforms for aaRS isoforms have been reported, including assays for *Pf*cProRS and *Hs*ProRS, they collectively suffer from several shortcomings that greatly limit sensitivity, robustness, and throughput. Particularly, the high enzyme concentrations that are required for these assays render them incapable of accurately profiling potent inhibitors. Non-radioactive aaRS assays generally require 0.1–0.5 μM enzyme and are consequently incapable of accurately measuring $K_D$-values substantially below this concentration range[22–25]. Additionally, current assay platforms require long incubation times and multiple manipulation steps that increase variability, largely preclude the measurement of binding kinetics, and are generally challenging to implement in high-throughput screening (HTS) settings. Therefore, a straightforward, single-step biochemical assay that facilitates HTS and reliable ligand characterization, including kinetic and substrate-dependent profiling, promises to greatly accelerate inhibitor development for this enzyme family.

Here, we show the development of a set of highly potent inhibitors, including previously elusive aaRS triple-site inhibitors that simultaneously target the adenosine, amino acid, and tRNA binding sites. Some analogs display potent and selective cellular activity against ABS and liver stage *Plasmodium* parasites and exhibit low propensity to resistance evolution. Supported by the development of a sensitive single-step biochemical assay for host and parasite ProRS that overcomes the limitations of existing assay platforms and co-crystal structures of representative inhibitor-target complexes, our work establishes a path forward for the rational development of selective next-generation malaria therapies.

## Results

The active site of ProRS comprises three distinct pockets that bind ATP, proline, and the 3′-terminal adenosine residue of tRNA^Pro (A76), respectively (Fig. 1a). Recently, Takeda Pharmaceuticals disclosed a new class of *Hs*ProRS inhibitors for the treatment of fibrosis[21,22,26]. Unlike halofuginone and analogs, which span the A76 and proline-binding sites and interact in an ATP-uncompetitive manner (i.e., the inhibitor affinity increases with increasing ATP concentration), this inhibitor class targets the ATP-binding pocket and features adjacent to the active site. Notably, some of these ligands, including T-3767758 (**2**) (Fig. 1a), displayed proline-uncompetitive steady state kinetics for *Hs*ProRS. We hypothesized that this property could be highly desirable for halofuginone-tolerant strains with elevated intracellular proline (halofuginone-induced), as it would potentially allow overcoming or even selecting against this resistance mechanism. Sequence analysis of *Hs*ProRS and *Pf*cProRS paralogs reveals overall high homology with several nonconserved residues in and adjacent to the binding site occupied by compound **2**, suggesting that inhibitors derived from this

series may be active against *Pf*cProRS and offer potential for the development of selective inhibitors (Supplementary Fig. 1)[8].

Compound **2** and analogs were readily accessible via a concise synthetic strategy starting from 3-aminopyrazine-2-carboxylic acid (see Supplementary Information). We tested the activity of compound **2** against ABS *P. falciparum*, which displayed good activity (EC$_{50}$ = 595 nM) against Dd2-2D4 wild-type parasites. Importantly, compound **2** did not show meaningful cross-resistance in halofuginone-induced parasites (EC$_{50}$ = 1.25 μM) and HFGR-I parasites (EC$_{50}$ = 736 nM), which, in addition to elevated proline, also feature the *Pf*cProRS^L482H mutation that renders them ~400-fold less sensitive to halofuginone (Fig. 1b)[8,27]. Encouraged by these results, we evaluated the reported co-crystal structure of compound **2** bound to *Hs*ProRS (PDB: 5VAD) more closely and noted that the cyclohexyl substituent adopts an unfavorable axial geometry (Fig. 1c). We reasoned that replacement of the cyclohexyl substituent with a piperidyl ring (NCP26, **3**) or addition of a geminal methyl group (compound **4**) would be energetically more favorable and would result in increased potency (Fig. 1d). Indeed, both compounds exhibited substantially increased activity against *P. falciparum* Dd2-2D4 (NCP26 EC$_{50}$ = 67.4 nM and compound **4** EC$_{50}$ = 180 nM), while retaining comparable potency against halofuginone-induced (NCP26 EC$_{50}$ = 120 nM and compound **4** EC$_{50}$ = 540 nM) and HFGR-I (NCP26 EC$_{50}$ = 68.6 nM and compound **4** EC$_{50}$ = 290 nM) parasites (Fig. 1b and Supplementary Fig. 2a). Moreover, unlike halofuginone, NCP26 did not induce rapid resistance in wild-type parasites that were subjected to intermittent drug pressure.

To further validate our hypothesis and support rational development efforts, we determined the co-crystal structure of NCP26 (Fig. 1e, PDB: 6T7K, and Supplementary Table 1) in complex with *Pf*cProRS. Notably, crystallization efforts were only successful in the presence of proline. As predicted, the piperidyl substituent of NCP26 adopts an energetically more favorable geometry compared to compound **2**. Furthermore, the formal cyclohexyl to piperidine substitution eliminates a potential stereocenter, simplifying future analog design.

Expanding on these findings with the goal to establish preliminary structure-activity relationships, we synthesized a targeted library of >20 analogs, including those exploring modifications to the 4-position of the cyclohexyl/piperidinyl moiety (Fig. 1d and Supplementary Fig. 3). We initially focused on this portion of the molecule since it would also facilitate the design of tracer ligands and hybrid molecules (see below). Activity profiling of this inhibitor set against wild-type ABS *P. falciparum* revealed a wide range of potencies, including several additional inhibitors with nanomolar potency. Although we did not identify an analog that improved upon the activity of NCP26, we found that the introduction of a 4-Boc-amino group to compound **2** and NCP26 (compounds **6** and **7**, respectively) was well tolerated, yielding inhibitors with comparable activity in wild-type parasites (Fig. 1d, Table 1, and Supplementary Fig. 2b–d, g). However, unlike the parent compounds, compounds **6** and **7** exhibited significantly decreased activity in halofuginone-induced parasites (Supplementary Fig. 2e). Interestingly, the activity was not further reduced in the HFGR-I line (Supplementary Fig. 2f), which is consistent with a proline-competitive binding mode, but no direct interaction within the proline-binding pocket[8].

### TR-FRET ProRS assay development and inhibitor characterization

While the pyrazinamide-series represented a promising chemotype for further ligand optimization, we recognized that rational development efforts would critically depend on our ability to reliably characterize the biochemical activity, including the substrate-dependent mode of inhibition, of analogs for *Pf*cProRS and *Hs*ProRS. As detailed above, current aaRS assay platforms suffer from several inherent limitations that render the accurate determination of binding affinities, binding kinetics, and mode of inhibition difficult. These constraints are

**Table 1 | Biochemical and in vitro activity of selected compounds**

| Compound | Dd2 ($EC_{50}$ [nM]) | | | $Pfc$ProRS ($K_D$ [nM]) | | | $Hs$ProRS ($K_D$ [nM]) | | |
|---|---|---|---|---|---|---|---|---|---|
| | wt | Halofuginone-induced | HFGR-I | – | + proline | + ATP | – | + proline | + ATP |
| Halofuginone (**1**) | 0.519 | 19.2 | 196 | >3000 | >3000 | 0.503 | 1160 | 2044 | 0.225 |
| **2** | 595 | 1250 | 736 | 437 | 7.17 | 291 | 371 | 2.16 | 954 |
| NCP26 (**3**) | 67.4 | 120 | 68.6 | 130 | 2.52 | 103 | 271 | 0.351 | 678 |
| **4** | 180 | 540 | 290 | 275 | 3.42 | – | 221 | 2.55 | – |
| **6** | 508 | 7930 | 6870 | 31.5 | 26.3 | – | 56.8 | 82.7 | – |
| **7** | 90.6 | 2170 | 2750 | 13.1 | 13.2 | – | 22.1 | 34.8 | – |
| MAT379 (**24**) | – | – | – | 100 | – | – | 1700 | – | – |
| ProSA (**25**) | 151 | 137 | 170 | 0.0559 | <0.5 | <0.5 | 0.0702 | <1 | <1 |
| Glyburide (**28**) | 75,600 | 73,600 | 69,600 | 2210 | 1020 | >15,000 | 73,900 | >50,000 | 30,900 |
| MAT334 (**29**) | 76%[b] | 59%[b] | 66%[b] | 1380 | 660 | – | 2110 | 2130 | – |
| MAT345 (**30**) | 249 | 7210 | 7010 | 39.8 | 35.9 | – | 140 | 282 | – |
| **31**[a] | 102%[b] | 98%[b] | 103%[b] | 2300 | 1990 | – | 5130 | >5000 | – |
| **32** | 63.2 | 1530 | 2730 | 6.67 | 5.35 | – | 72.1 | 270 | – |
| MAT436 (**34**) | 6.76 | 251 | 2630 | 0.375 | 0.722 | – | 0.465 | 6.65 | – |
| **35** | 18.7 | 355 | 4990 | 0.680 | 0.729 | – | 0.35 | 3.49 | – |
| **36** | 18.7 | 383 | 5730 | 0.821 | 0.772 | – | 0.367 | 6.63 | – |
| **37** | 59%[b] | 7150 | 6950 | 246 | 175 | – | 502 | 570 | – |
| **38** | 3950 | 6710 | 5590 | 455 | 279 | – | 720 | 609 | – |
| **39** | 72%[c] | 98%[c] | 95%[c] | 227 | 179 | – | 216 | 658 | – |
| *iso*-MAT436 (**40**) | 21.9 | 979 | 6820 | 4.57 | 3.91 | – | 18.6 | 20.8 | – |

TR-FRET data are presented as the mean of $n \geq 2$ independent replicate wells and are representative of $n \geq 3$ independent experiments. Parasite (Dd2) growth data are presented as the mean of $n \geq 2$ biological replicates and are representative of $n \geq 3$ independent experiments.

*wt* wild-type (Dd2-2D4).

[a]Compound **31** was only tested in one independent experiment ($n = 6$ biological replicates) against each parasite strain.

[b]Data shown is percent growth at 10 µM because compound had <50% inhibition (>50% growth).

[c]Data shown is percent growth at 1 µM because compound had <50% inhibition (>50% growth). Complete dataset including all compounds and statistical analysis is available in Source Data—Inhibitor Profiling.

particularly relevant for potent ligands, which existing assays cannot differentiate or quantitatively measure their affinity values.

To this end, we envisioned that the favorable characteristics of time-resolved Förster resonance energy transfer (TR-FRET) assays, including high sensitivity, specificity, and flexibility, would offer an equally straightforward and robust platform for the quantitative characterization of aaRS ligands (Fig. 2a)[28–32]. However, no suitable fluorescently-labeled tracers for TR-FRET (or analogous fluorescence polarization) assays have been reported for ProRS or other aaRS enzymes. This is likely because the aaRS active site is generally deeply buried in the ligand-bound state, which renders the development of linker-modified ligands challenging[15].

Informed by our crystallographic data and the good cellular activity of pyrazinamide **7**, we hypothesized that the 4-amino-piperidyl substituent represented a suitable position for linker functionalization, serving as a starting point in the design of a fluorescent tracer for TR-FRET-based ligand displacement assays (Fig. 2a). Replacement of the Boc-group with an acyl linker should follow the triphosphate exit vector, as in the reported halofuginone-ATP *Pfc*ProRS co-crystal structures (e.g., PDB: 4OLF). Based on these insights, we synthesized and evaluated several TR-FRET tracers, including MAT379 (**24**) and MAT425 (Fig. 2b), that were appropriate for the development of a single-step ligand displacement assay, enabling screening of active site inhibitors for ProRS.

We cloned a codon-optimized construct and recombinantly expressed the ProRS domain of *Pfc*ProRS (aa249–746)[24] as an *N*-terminal His6-HaloTag fusion protein (HT-*Pfc*ProRS). The HaloTag is a self-labeling protein tag that allows for efficient and defined covalent attachment of HaloTag-ligand modified small molecules, which we exploited to functionalize HT-*Pfc*ProRS with CoraFluor-1-Halo as the TR-FRET donor[32].

Saturation binding studies with CoraFluor-1-labeled HT-*Pfc*ProRS (1 nM) and MAT379 or MAT425 showed a dose-dependent increase in TR-FRET signal consistent with a specific, one-site, monophasic association model (Fig. 2c and Supplementary Fig. 4a). Non-linear regression analysis yielded equilibrium dissociation constants ($K_D$) of 100 nM and 199 nM, respectively. We selected MAT379 for further characterization because of its higher affinity. To determine the minimum incubation time required for equilibrium conditions (five half-lives), we measured the first-order dissociation rate constant ($k_{off}$) for MAT379 by 10-fold dilution of an equilibrated solution of CoraFluor-1-labeled HT-*Pfc*ProRS (100 nM) and -$EC_{80}$ MAT379 (560 nM) which yielded a $k_{off}$-value of <0.16 $min^{-1}$ (Supplementary Fig. 4c, d), suggesting that quasi-equilibrium is reached within 15 min (unless the test compounds themselves exhibit slow binding kinetics)[33].

Dose-titration of ATP and proline revealed that MAT379 is not only ATP-competitive, as expected, but, unlike compound **2** or NCP26, also proline-competitive, which is consistent with our finding that compound **7** was less active in halofuginone-induced and HFGR-I parasites (Supplementary Fig. 2c). Quantification of the individual binding affinities yielded $K_D$-values for ATP ($K_D = 892$ µM) and proline ($K_D = 457$ µM) (Supplementary Fig. 5). This outcome is ideal since it allows for highly sensitive, direct determination of both substrate-dependence and inhibitor-competitiveness with respect to proline and ATP, offering a key advantage over other assay platforms.

To further explore the scope and limitations of our TR-FRET assay approach, we next used a reference compound set comprising halofuginone (proline-competitive and ATP-uncompetitive), NCP26 (proline-uncompetitive and ATP-competitive), and the non-hydrolyzable prolyl-AMP analog ProSA (**25**) (proline- and ATP-competitive) (Fig. 2b). Dose-response titration of the test compounds were performed using 250 nM MAT379 as tracer (2.5x $K_D$) in the absence or presence of

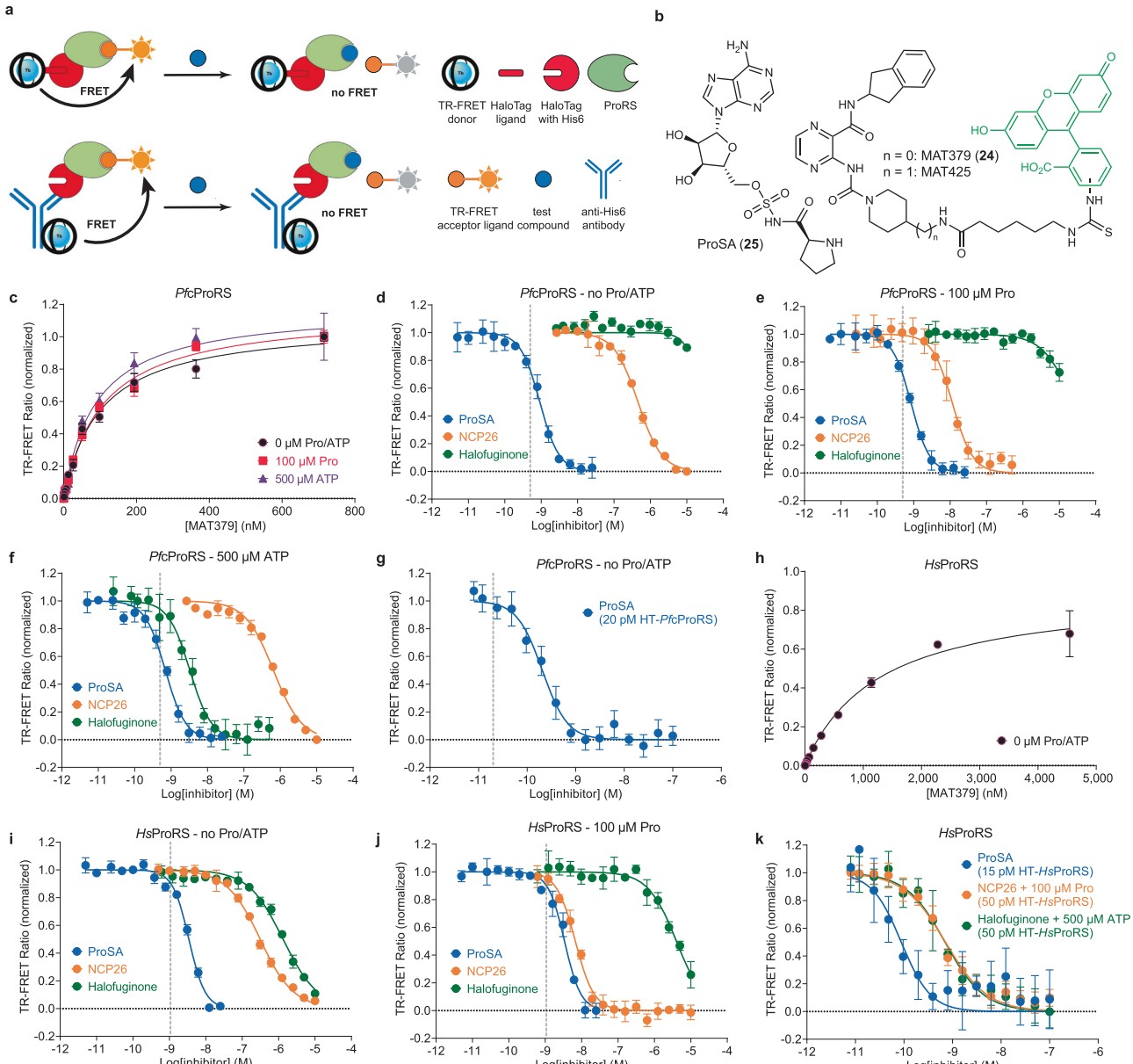

**Fig. 2 | TR-FRET-based ProRS assay design and validation. a** Principle of His6-HaloTag-ProRS (HT-ProRS) ligand displacement assay. The TR-FRET donor is installed either via labeling of the HaloTag with CoraFluor-1-functionalized HaloTag ligand (CoraFluor-1-Halo) and/or using a CoraFluor-1-labeled anti-His6 antibody (CoraFluor-1-Pfp)[32]. Positive TR-FRET signal is observed upon binding of a suitable tracer labeled with a compatible fluorescence acceptor. Displacement of the tracer by a test compound disrupts the signal. **b** Chemical structures of ProSA (**25**) and tracers MAT379 (**24**) and MAT425. **c** Saturation binding of MAT379 to CoraFluor-1-labeled HT-*Pfc*ProRS (1 nM) in the absence (black) or presence of 100 μM Pro (red) or 500 μM ATP (purple). TR-FRET ratios were background-corrected relative to 10 μM ProSA (~20,000 × $K_D$). Dose-response titration of reference compounds (ProSA (blue), NCP26 (orange), halofuginone (green)) using CoraFluor-1-labeled HT-*Pfc*ProRS (0.25–1 nM) and MAT379 at 2.5x $K_D$ (250 nM) in the absence (**d**) or presence of 100 μM Pro (**e**) or 500 μM ATP (**f**). Under all conditions, ProSA is titrating HT-*Pfc*ProRS. **g** Dose-response

titration of ProSA using CoraFluor-1-labeled HT-*Pfc*ProRS (20 pM), CoraFluor-1-labeled anti-His6 antibody (1 nM), and MAT379 at 2.5x $K_D$ (250 nM). **h** Saturation binding of MAT379 to CoraFluor-1-labeled HT-*Hs*ProRS (1.5 nM). TR-FRET ratios were background-corrected relative to 10 μM ProSA (~20,000 × $K_D$). Dose-response titration of reference compounds using CoraFluor-1-labeled HT-*Hs*ProRS (1 nM) and MAT379 as tracer at 0.15x $K_D$ (250 nM) in the absence (**i**) or presence of 100 μM Pro (**j**). ProSA is titrating HT-*Hs*ProRS in both conditions and NCP26 is titrating HT-*Hs*ProRS in the presence of 100 μM Pro. **k** Dose-response titration of test compounds using indicated concentrations CoraFluor-1-labeled HT-*Hs*ProRS, 1 nM CoraFluor-1-labeled anti-His6 antibody, and MAT379 as tracer at 0.15x $K_D$ (250 nM). Data are expressed as mean ± SD ($n$ = 3 independent replicate wells, except for NCP26 in **d** and **f** where $n$ = 2, for halofuginone in **e** where $n$ = 2, and for everything in **i** and **j** where $n$ = 6) and are representative of at least two independent experiments. Gray vertical dotted line indicates calculated titration limit.

100 μM proline or 500 μM ATP. As shown in Fig. 2d–f, NCP26 exhibited a 52-fold increased affinity in the presence of proline, consistent with the predicted proline-uncompetitive mode of inhibition ($K_{D,0\ \mu M\ proline}$ = 130 nM vs. $K_{D,100\ \mu M\ proline}$ = 2.52 nM). By contrast, halofuginone displayed ATP-uncompetitive binding, showing a >6000-fold increased affinity in the presence of ATP

($K_{D,0\ \mu M\ ATP}$ > 3 μM vs. $K_{D,500\ \mu M\ ATP}$ = 503 pM). However, the high affinity of ProSA resulted in titrating the enzyme ($K_D$ < 0.5 nM; [HT-*Pfc*ProRS] = 1 nM) and limited our ability to obtain accurate equilibrium dissociation constants under these conditions.

To further improve the sensitivity of our assay for high-affinity ligands, we explored an alternative labeling strategy employing a

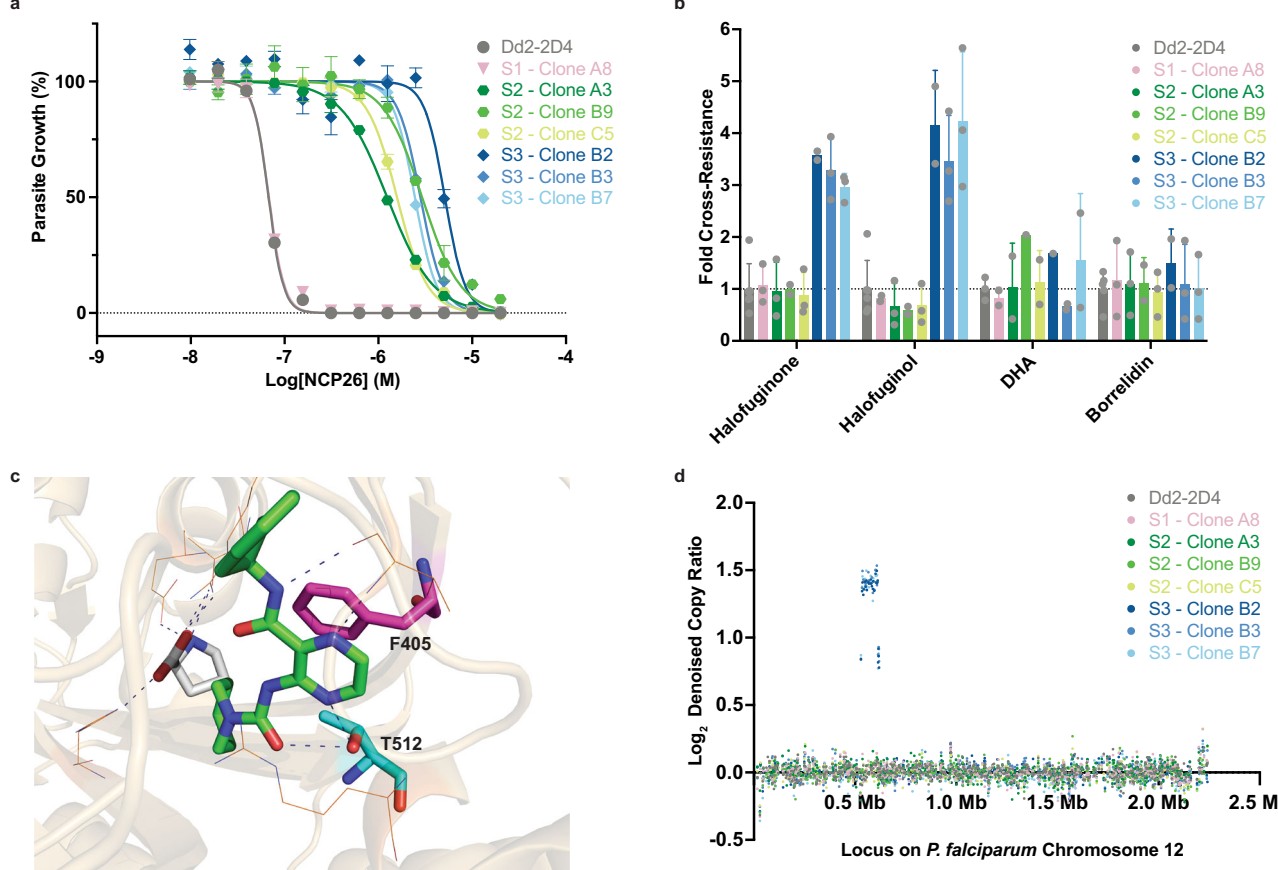

**Fig. 3 | NCP26-resistance selection and whole genome sequence analysis. a** In vitro activity of NCP26 against *ABS P. falciparum* Dd2-2D4 (parent) or subclones from three independent resistance selection experiments (S1-3). S1 did not yield resistant parasites and S1−clone A8 was included for comparison. **b** In vitro activity of reference compounds against Dd2-2D4 (parent) or subclones from each selection (S1-3). Reference compounds include *Pf*cProRS inhibitors (halofuginone and halofuginol) and non-*Pf*cProRS inhibitors dihydroartemisinin (DHA) and borrelidin (*P. falciparum* threonyl-tRNA synthetase inhibitor). **c** Residues T512S (S2) and F405L (S3) were identified by whole genome sequence analysis to mediate NCP26-resistance and are mapped to the co-crystal structure of NCP26 (green) and proline (white) bound to wild-type *Pf*cProRS (PDB: 6T7K). Both amino acid side chains directly interact with the pyrazinamide core. **d** Whole genome sequence analysis revealed amplification

of the intra-chromosomal region harboring the *Pf*cProRS locus on chromosome 12 that is observed in all S3 clones. No copy number variations were observed in other chromosomes for S3 clones or in any chromosomes for Dd2-2D4 (parent), S1 (no NCP26-resistance observed), or S2 (*Pf*cProRS$^{T512S}$). Detailed CNV data are available in Source Data−CNV. ABS growth assay data in **a** are expressed as mean ± s.d. (*n* = 3 biological replicates) and are representative of at least two independent experiments. ABS growth assay data in **b** are expressed as the mean IC$_{50}$ values from independent experiments ± s.d. (*n* = 3 biological replicates per independent experiment) and include at least two independent experiments, except for DHA in S2−clone B9 which only was only tested once. For **a**, **b**, and **d**, clonal parasite lines isolated from the same selection flask are shown as different shades of the same color (S1 = pink, S2 = shades of green, and S3 = shades of blue).

CoraFluor-1-labeled anti-His6 antibody, which enabled the installation of multiple TR-FRET donors in addition to the donor covalently attached to the HaloTag label[32]. This approach allowed us to perform the assay with as little as 20 pM HT-*Pf*cProRS, which is >1000 and >100-fold lower than current non-radioactive and radioactive assay platforms, respectively. This strategy enabled the determination of an accurate $K_D$-value for ProSA ($K_D$ = 55.9 pM) (Fig. 2g) which, to the best of our knowledge, has not been reported previously.

Although the direct labeling of HT-*Pf*cProRS with CoraFluor-1-Halo was less sensitive than the CoraFluor-1-labeled antibody-based strategy, it was still sufficient for most applications and was therefore selected as the default platform because of its simplicity and excellent robustness (Z′ > 0.95 at 500 pM HT-*Pf*cProRS, Z′ = 0.71 at 250 pM HT-*Pf*cProRS, and Z′ = 0.60 at 20 pM HT-*Pf*cProRS supplemented with 1 nM CoraFluor-1-labeled anti-His6 antibody)[34].

Following the same approach, we adapted and optimized the assay for *Hs*ProRS to enable quantitative comparative biochemical profiling and, as an added benefit, facilitate discovery efforts aiming to target the human paralog directly. As for *Pf*cProRS, we expressed

*Hs*ProRS (aa996−1512) as an *N*-terminal His6-HaloTag fusion protein (HT-*Hs*ProRS), which was readily labeled with CoraFluor-1-Halo. Dose-titration of MAT379 and MAT425 revealed that the respective tracers bound to HT-*Hs*ProRS 17-fold less tightly ($K_{D,MAT379}$ = 1.70 μM) and >50-fold less tightly ($K_{D,MAT425}$ > 10 μM) relative to HT-*Pf*cProRS, establishing MAT379 as the preferred tracer for both ProRS paralogs (Fig. 2h and Supplementary Fig. 4b). Consistent with reduced affinity of MAT379 for HT-*Hs*ProRS compared to HT-*Pf*cProRS, the measured first-order dissociation rate was faster, and the calculated first-order association rate was slower (Supplementary Fig. 4c, d).

Although MAT379 exhibited lower affinity for HT-*Hs*ProRS, the tracer was still well suited for assay development. The slight decrease in signal intensity was readily compensated by increased enzyme concentration to yield an equally robust assay (Z′ = 0.80 at 1 nM CoraFluor-1-labeled HT-*Hs*ProRS and Z′ = 0.76 at 50 pM CoraFluor-1-labeled HT-*Hs*ProRS supplemented with 1 nM CoraFluor-1-labeled anti-His6 antibody). Similar to HT-*Pf*cProRS, we found MAT379 binding to HT-*Hs*ProRS to be both ATP ($K_D$ = 30.6 μM) and proline ($K_D$ = 67.1 μM) competitive. We next determined the $K_D$-values of our reference

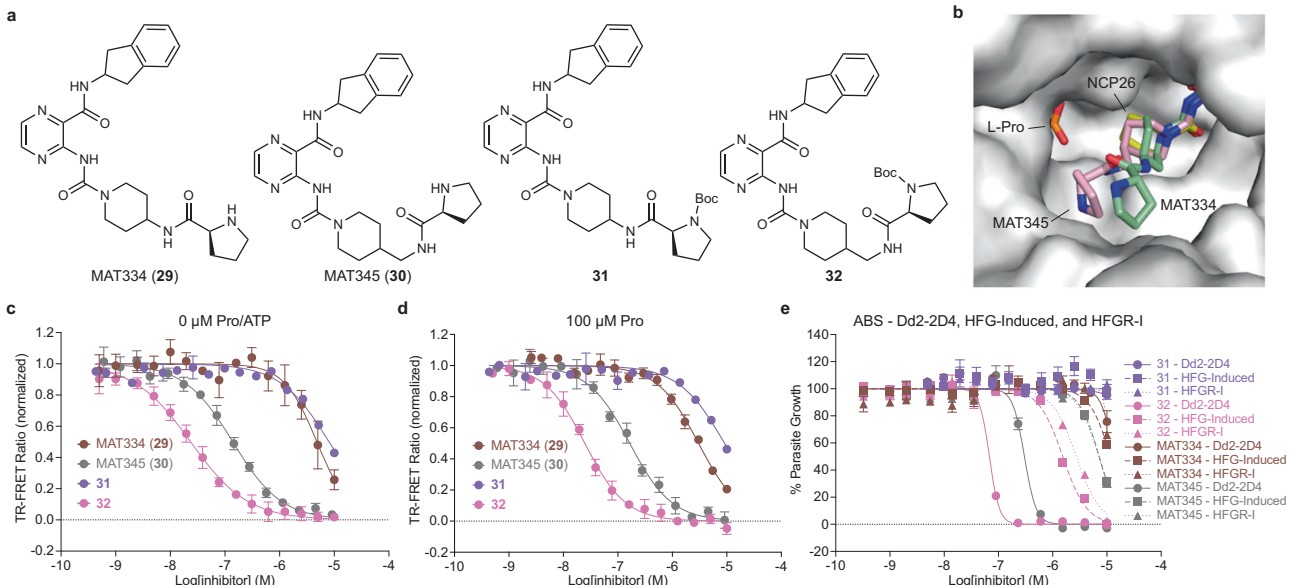

**Fig. 4 | Dual-site ProRS inhibitors. a** Chemical structures of pyrazinamide-proline hybrids (absolute stereochemistry). **b** Overlay of the co-crystal structures of *Pf*cProRS (gray surface) in complex with proline (orange sticks) and either NCP26 (PDB: 6T7K, yellow sticks), MAT334 (**29**) (PDB: 7QC2, green sticks), and MAT345 (**30**) (PDB: 7QB7, pink sticks) shows the prolyl-substituents of MAT334 and MAT345 pointing outside the active site. Dose-response titration of pyrazinamide-proline hybrids using CoraFluor-1-labeled HT-*Pf*cProRS (0.25–1 nM) and MAT379 as tracer at 2.5x $K_D$ (250 nM) in the absence (**c**) or presence of 100 µM Pro (**d**). See Supplementary Fig. 17 for corresponding data with HT-*Hs*ProRS in the presence or absence of 100 µM Pro. **e** In vitro activity of pyrazinamide-proline hybrids against wild-type (Dd2-2D4, circles and solid lines), halofuginone-induced (HFG-induced, squares and dashed lines), and HFGR-I (triangles and dotted lines) *P. falciparum* asexual blood stage. TR-FRET assay data in **c** and **d** are expressed as mean ± s.d. (*n* = 3 independent replicate wells, except for compound **31** where *n* = 2) and are representative of at least two independent experiments. ABS growth assay data in **e** are expressed as mean ± s.d. (*n* = 3 biological replicates) and are representative of at least three independent experiments, except for compound **31** which was tested once for each strain shown. In **c**–**e**, MAT334 is shown in brown, MAT345 in gray, **31** in purple, and **32** in pink.

inhibitor set in the presence and absence of substrates for *Hs*ProRS (Fig. 2i–k). Interestingly, NCP26 was marginally selective for *Pf*cProRS in the absence of substrates but displayed >750-fold increased affinity for *Hs*ProRS in the presence of 100 µM proline (compared to 52-fold for *Pf*cProRS), yielding a $K_D$-value of 351 pM for the host enzyme, which is comparable to the affinity of halofuginone in the presence of 500 µM ATP ($K_D$ = 225 pM).

Following the comprehensive validation of our TR-FRET assay platform, we profiled the complete set of pyrazinamide analogs together with several reference compounds such as halofuginol (**26**), D-ProSA (**27**) as a negative control for ProSA, and glyburide (**28**) (Supplementary Fig. 3). We first tested the inhibitor set in a dose-response format against both CoraFluor-1-Halo-labeled HT-*Pf*cProRS and HT-*Hs*ProRS, in the absence and presence of individual substrates, to determine the quantitative binding affinities and modes of inhibition. Inhibitors that exhibited ligand depletion under the default assay conditions were retested at lower ProRS concentrations using our antibody-based labeling protocol (Supplementary Figs. 6–15).

Most pyrazinamides were unselective or displayed modest preference for *Hs*ProRS (Supplementary Fig. 16a). Overall, the relative biochemical affinity of the pyrazinamide series tracked well with the cell-based activity, with proline-uncompetitive inhibitors showing little to no differential activity in wild-type, halofuginone-induced, and HFGR-I parasite strains, while proline-competitive compounds exhibited substantial cross-resistance, consistent with on-target activity (Table 1, Supplementary Fig. 16b, and Source Data−Inhibitor Profiling).

We also confirmed that glyburide, which has previously been identified as a parasite-selective inhibitor that targets *Pf*cProRS allosterically adjacent to the active site, displayed >30-fold selectivity in the absence of substrates and, consistent with the original report, was ATP- and proline-competitive (Supplementary Figs. 6–15, Table 1, and Source Data−Inhibitor Profiling)[24].

## Validation of pyrazinamide mode of action

The strong correlation ($r_s$ = 0.85) between the biochemical activity of this inhibitor set and the cellular potency against ABS *P. falciparum* strongly suggests that the antiparasitic activity of pyrazinamide analogs is the direct consequence of on-target activity. To experimentally validate *Pf*cProRS as the principle functional target and to further assess the propensity of resistance evolution, we conducted three independent selections (S1-3) under intermittent drug pressure with NCP26 in ABS *P. falciparum* Dd2-2D4 parasites in vitro[35–38]. Unlike halofuginone, which yields resistant parasites within <5 generations, we observed moderate resistance (~20-to-80-fold) under NCP26 drug pressure only in large scale experiments ($10^9$ parasites) and prolonged culture periods (>50 generations, 100 days) in 2 out of 3 selections. We next cloned parasites from all three selections, including S1, which failed to yield resistant bulk parasites, as an additional reference. Like the bulk populations, the clonal lines exhibited the same level of NCP26-resistance (Fig. 3a and Source Data−NCP26 Resistance Selection), but no (S1-2) or low-level (<5-fold, S3) cross-resistance to halofuginone analogs and no differential sensitivity to other drugs, such as dihydroartemisinin (DHA) or the threonyl-tRNA synthetase inhibitor borrelidin (Fig. 3b and Source Data−NCP26 Resistance Selection).

Whole genome analysis of the individual NCP26-resistant parasite clones identified two independent single amino acid changes, *Pf*cProRS[F405L] in S2 and *Pf*cProRS[T512S] in S3, mapping to the adenosine binding pocket, which form direct contacts with the pyrazinamide core (Fig. 3c and Source Data−Mutation Calls). Notably, while no copy number variation (CNV) was observed in parasites carrying the *Pf*cProRS[T512S] mutation, *Pf*cProRS[F405L] clones had a ~3-fold amplification of the intra-chromosomal region harboring the *Pf*cProRS locus, with one mutant and two wild-type alleles, which is also consistent with the slightly reduced activity of halofuginone toward these parasites (Fig. 3d and Source Data−CNV). Together, these results establish

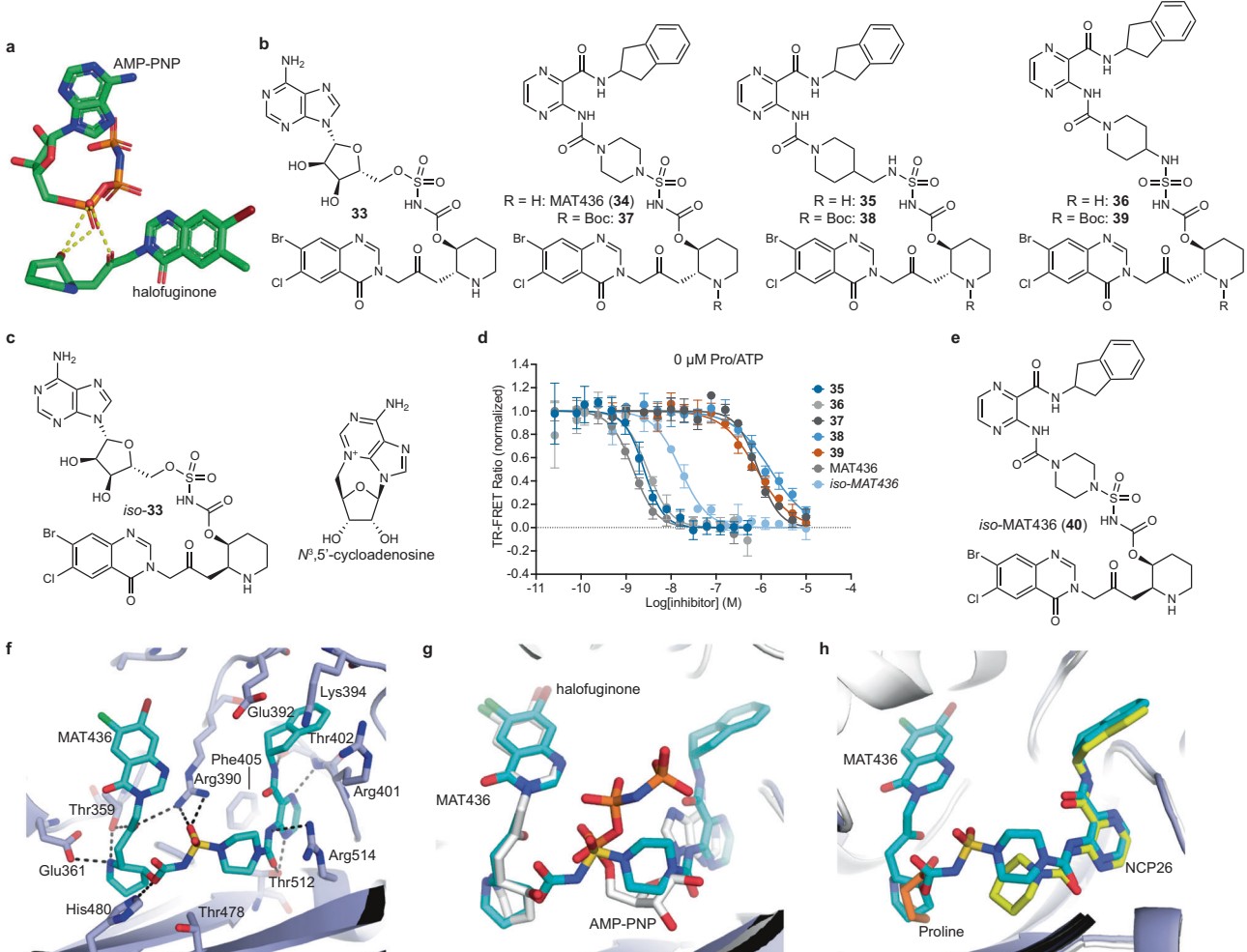

**Fig. 5 | Triple-site ProRS inhibitors biochemical characterization. a** Interaction of halofuginone with non-hydrolyzable ATP-analog AMP-PNP bound to *Hs*ProRS (PDB: 4HVC). Hydrogen-bonds are indicated by yellow dashed lines. **b** Chemical structures of triple-site inhibitors based on halofugione-AMP and pyrazinamide-halofuginone hybrid design, with relative stereochemistry shown for halofuginone moiety. **c** Chemical structures of *iso*-33 (relative stereochemistry for halofuginone moiety) and cyclization byproduct $N^3$,5′-cycloadenosine. **d** Dose-response titration of pyrazinamide-halofuginone hybrids using CoraFluor-1-labeled HT-*Pf*cProRS (0.25−0.5 nM) and MAT379 as tracer at 2.5x $K_D$ (250 nM). Inhibitors are color coded: **35** (dark blue), **36** (light gray), **37** (dark gray), **38** (blue), **39** (red), MAT436 (gray), and *iso*-MAT436 (light blue). See Supplementary Fig. 18 for corresponding data in the presence of 100 µM Pro and with HT-*Hs*ProRS in the presence or absence of

100 µM Pro. **e** Chemical structure of *iso*-MAT436 (**40**) with relative stereochemistry for halofuginone moiety. **f** Co-crystal structure of MAT436 (cyan sticks, PDB: 7QC1) bound to *Pf*cProRS reveals electrostatic interactions (dashed lines) with conserved residues within the active site (light blue sticks) and occupancy of all three substrate-binding pockets in the active site. Overlay of co-crystal structures of *Pf*cProRS in complex with MAT436 (cyan sticks, PDB: 7QC1) with either (**g**) *Pf*cProRS in complex with halofuginone and AMP-PNP (silver sticks, PDB: 4Q15), and (**h**) *Pf*cProRS in complex with NCP26 (yellow sticks) and proline (orange sticks, PDB: 6T7K). TR-FRET assay data in **d** are expressed as mean ± s.d. (for **37**, **38**, **39**, and *iso*-MAT436: *n* = 2 independent replicate wells; for MAT436, **35**, and **36**: *n* = 6) and are representative of at least two independent experiments.

---

strong evidence that *Pf*cProRS constitutes the mechanistic target of NCP26.

## Development of multi-site inhibitors

Following validation of the 4-position of NCP26 for derivatization, we next explored the suitability of the pyrazinamide scaffold for the design of dual- and triple-site inhibitors that extend to the proline- and tRNA^Pro A76-binding pockets, mimicking ProSA or the halofuginone/ ATP complex, respectively. Such compounds have been hypothesized to bind with very high affinity, while retaining the ability for the development of species-selective ligands[39].

We first designed a series of formal pyrazinamide-proline hybrids, MAT334 (**29**) and MAT345 (**30**), structurally analogous to ProSA, aiming to identify a suitable linker between the two binding sites (Fig. 4a). Our preliminary docking studies suggested 4-amino and 4-aminomethyl spacers to be appropriate. Biochemical profiling of both compounds, including their Boc-protected precursors (**31** and

**32**), against *Pf*cProRS revealed a stark difference in potency between the two inhibitor pairs, identifying compounds **30/32** as the most potent pair with equivalent potency in the absence and presence of proline (Fig. 4b, c). Unexpectedly, **32** ($K_D$ = 6.67 nM) was substantially higher affinity than the corresponding deprotected target compound MAT345 ($K_D$ = 39.8 nM). The poor biochemical activity of proline-hybrids **29/31** was recapitulated by the lack of activity against ABS *P. falciparum*, whereas compound **32**, which was of comparable potency to NCP26, and MAT345 inhibited parasite growth with $EC_{50,Dd2-2D4}$ = 63.2 nM and 249 nM, respectively (Fig. 4d). Like compound **7**, the activity in halofuginone-induced and HFGR-I parasites was 20−40-fold decreased. Notably, we were only able to obtain co-crystal structures for compounds MAT334 and MAT345 with *Pf*cProRS in the presence, but not in the absence, of proline (PDB: 7QC2 and 7QB7, respectively, and Supplementary Table 1). Analysis of both co-crystal structures revealed that proline occupies the amino acid binding pocket, while the prolyl-substituent of MAT334/345 is pointing outside the active

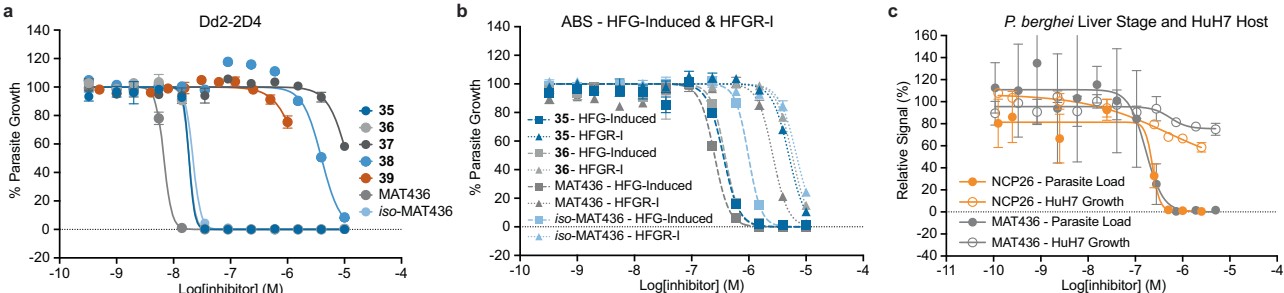

**Fig. 6 | Cellular activities of triple-site ProRS inhibitors. a** In vitro activity of pyrazinamide-halofuginone hybrids and Boc-protected precursors against wild-type ABS *P. falciparum* Dd2-2D4. **b** In vitro activity of pyrazinamide-halofuginone hybrids against halofuginone-induced (HFG-Induced, squares and dashed lines) and HFGR-I (triangles and dotted lines) ABS *P. falciparum*. **c** In vitro activity of NCP26 and MAT436 against liver stage *P. berghei* (solid circles) and HuH7 liver cells (open circles). In **a**–**c**, inhibitors are color coded: **35** (dark blue), **36** (light gray), **37** (dark gray), **38** (blue), **39** (red), MAT436 (gray), *iso*-MAT436 (light blue), and NCP26 (orange). ABS growth assay data in **a** and **b** are expressed as mean ± s.d. (*n* = 3 biological replicates) and are representative of at least three independent experiments, except for compound **39** which is only representative of two independent experiments for Dd2-2D4 in **a**. Liver stage and HuH7 data in **c** are expressed as mean ± s.d. (*n* = 3 biological replicates) and are representative of at least three independent experiments.

site and does not form discernible electrostatic interactions with surrounding residues, which suggests the prolyl moiety is highly flexible in this binding mode (Fig. 4b). While unexpected, these results are not inconsistent with a lack of proline-uncompetitive binding as determined in our biochemical assay, and a proline-competitive mode of inhibition as determined by cellular inhibitor profiling (Fig. 4c–e and Supplementary Fig. 17) and do not demonstrate that the prolyl-substituent is unable to bind the amino acid pocket in the absence of proline, but rather show that crystallization efforts were unsuccessful in the absence of proline. Interestingly, the comparison of the crystal structures of apo (PDB: 4K86) and proline-bound (PDB: 7OSY) *Hs*ProRS shows a major reorientation of amino acid residues in the ATP-pocket upon proline binding, which closely resembles the proline and T-3767758 co-crystal structure (PDB: 5VAD), as well as the residues lining the active site rim, which interact with the prolyl-substituent of MAT334/345 (Supplementary Fig. 18). Together, these observations provide a rationale for the proline-uncompetitive binding of the simple pyrazinamide-based ligands (e.g., NCP26) and the proline-competitive binding of the modified analogs (e.g., MAT334 and MAT345).

Analysis of the co-crystal structure of halofuginone and ATP with ProRS illustrates the molecular basis of halofuginone's ATP-uncompetitive binding mode[15,24,40]. The ketone and hydroxyl-group of halofuginone form two well-defined hydrogen bonds with the α-phosphate of ATP, furnishing the intrinsic inhibitory complex in these ProRS ligand structures (Fig. 5a)[15,24]. The importance of ATP for halofuginone binding to *Pf*cProRS and *Hs*ProRS is confirmed by our biochemical data that show three orders of magnitude lower affinity in the absence of ATP (see Fig. 2). As indicated before, we hypothesized that covalently linking both fragments would greatly improve potency. We speculated that the quinazolinone moiety would anchor the halofuginone fragment in the A76-binding pocket and enable true triple-site inhibitors that bind all three substrate-binding sites[39].

As a proof-of-concept, we initially designed compound **33** as a formal hybrid of halofuginone and adenosine that joins both ligands via a sulfamoylcarbamate, analogous to ProSA (Fig. 5b). Unfortunately, compound **33** was prone to fragmentation by intramolecular cyclization to $N^3$,5′-cycloadenosine, which has been observed previously for aminoacyl-sulfamoyl-adenosine (aaSA) analogs, and the 2-position of the hydroxypiperidine was susceptible to epimerization to *iso*-**33** during purification, which was previously observed with halofuginone (Fig. 5c)[8,41,42]. We therefore synthesized a series of pyrazinamide-halofuginone hybrids exploring three different linker-elements, while retaining the sulfamidylcarbamate as an acylphosphate isostere. In addition to MAT436 (**34**), which our modeling studies suggested to be

optimal, we also synthesized the linker homologs **35** and **36** (Fig. 5b). Gratifyingly, all compounds exhibited very high affinities for *Pf*cProRS and *Hs*ProRS in the absence of ATP. As predicted, MAT436 was the highest affinity ligand and displayed high affinity for *Pf*cProRS ($K_D$ = 375 pM, >10⁴-fold better than halofuginone in the absence of ATP and ~350-fold better than NCP26 in the absence of proline), while compounds **35** ($K_D$ = 680 pM) and **36** ($K_D$ = 821 pM) were only marginally less potent (Fig. 5d and Supplementary Fig. 19). In contrast, the Boc-protected precursors (**37**, **38**, **39**), which cannot bind the proline-binding pocket because of steric constraints, were much weaker ligands for *Pf*cProRS, with $K_D$-values comparable to the prolyl-hybrids **29** and **30**, suggesting that these compounds can still bind with the halofuginone moiety pointing outside the active site. Notably, as for the adenosine-halofuginone hybrid **33**, we observed that MAT436 was susceptible to epimerization of the 2′-position of the piperidine under non-acidic conditions, resulting in the corresponding *syn*-substituted analog (*iso*-MAT436, **40**), which had only ~10-fold lower affinity (Fig. 5d, e and Table 1).

To provide further support for the predicted binding modes, we co-crystallized MAT436 with *Pf*cProRS and solved the structure to 2.51 Å resolution (PDB: 7QC1 and Supplementary Table 1). As illustrated in Fig. 5f, MAT436 occupies the ATP-, proline-, and A76-binding pockets, where it forms electrostatic interactions with conserved residues similar to halofuginone, proline, and NCP26. An overlay of MAT436 bound to *Pf*cProRS with the corresponding ternary complex of *Pf*cProRS with halofuginone and AMP-PNP (PDB: 4Q15) revealed a nearly perfect overlap with halofuginone, while the pyrazinamides of MAT436 and NCP26 (PDB: 6T7K) were virtually superimposable (Fig. 5g, h). These results unambiguously validate the pyrazinamide-halofuginone hybrids as true ProRS triple-site ligands that occupy all three substrate-binding sites.

We next evaluated the activity of the pyrazinamide-halofuginone hybrids against ABS *P. falciparum*. Importantly, the high affinity of **35**, **36**, and MAT436 translated well to cell-based assays. All three compounds inhibited *P. falciparum* ABS growth at low nanomolar concentrations with comparable potencies (MAT436, EC₅₀ = 6.8 nM; **35**, EC₅₀ = 18.7 nM; **36**, EC₅₀ = 18.7 nM), while the corresponding Boc-protected precursors (**37**, **38**, **39**) were, as expected, substantially less active (>1 μM) (Fig. 6a). Interestingly, the epimerized analog of MAT436, *iso*-MAT436 (**40**), was only 3-fold less active, which overall tracked well with its 5–10-fold reduced biochemical potency. Unfortunately, but not unexpected, MAT436, **35**, **36**, and **40** exhibited reduced activity toward halofuginone-induced and HFGR-I-mutant parasites, comparable to halofuginone (Fig. 6b). Although undesirable, this finding provides further validation for on-target activity.

## Liver stage and host toxicity evaluation

We have previously shown that halofuginone and derivatives are active against liver stage *Plasmodium* parasites[7,8]. To test whether our pyrazinamide-based inhibitors would exhibit comparable activity, we evaluated NCP26 and MAT436 in a tissue culture-based *P. berghei* liver stage model. Specifically, we added the various test compounds in dose-response to human hepatocytes (HuH7) followed by infection with luciferase-expressing *P. berghei* ANKA sporozoites, which enabled quantification of parasite load by luminescence. In parallel, we also assessed host cell viability to provide an estimate of cellular selectivity. Both compounds displayed potent antiparasitic activity resulting in complete inhibition of *P. berghei* growth at submicromolar concentrations (NCP26 $EC_{50} = 232$ nM and MAT436 $EC_{50} = 167$ nM), while showing less pronounced activity in human hepatocytes that suggested host cytostatic activity rather than the cytotoxic activity that is expected at higher concentrations (Fig. 6c). This is consistent with our own findings for halofuginone and halofuginol, which, despite comparable biochemical activity, exhibit selectivity for parasite vs. host activity in cell-based assays[8].

## Discussion

Motivated by the urgent need for next-generation antimalarial therapeutics, we have developed multiple orthogonal classes of *Pf*cProRS inhibitors and a generalizable high-throughput assay platform to support the comprehensive biochemical profiling of ProRS inhibitors against the parasite and human paralogs. Our TR-FRET-based ligand displacement assay strategy resolves the limitations of current platforms that have stymied aaRS-targeted drug development and offers high throughput, robustness, sensitivity, and flexibility. Our methodology is based on a simple mix-and-read assay design that enables kinetic measurements and detailed interrogation of inhibition modes, while reducing the required amount of protein by several orders of magnitude. These characteristics not only improve economic aspects, but, more importantly, allow for the quantitative profiling of high-affinity ligands and the accurate measurement of equilibrium binding constants, including for ProSA which, to the best of our knowledge, has not been reported before. The presented CoraFluor-ProRS technology promises to greatly accelerate the drug discovery process beyond malaria and will likely be equally applicable to other parasitic diseases where the corresponding ProRS homolog is a validated drug target, including toxoplasmosis, leishmaniasis, cryptosporidiosis, and coccidiosis[40]. Moreover, host aaRSs have been recognized for their many roles in human health and disease, and *Hs*ProRS has been proposed as an attractive target for the development of new drug classes for the treatment of autoimmune disorders, fibrosis, cancer, and more recently viral infections, including COVID-19, chikungunya, and dengue[12,13].

Furthermore, supported by our crystallographic data, we have developed inhibitor classes with high affinity for ProRS and potent dual-stage activity against ABS and liver stage *Plasmodium* parasites. These include a set of ATP-site directed, proline-uncompetitive inhibitors with low- to sub-nanomolar affinity that overcome halofuginone resistance mechanisms or other forms of rapid drug tolerance. The inhibitor design was inspired by a previous report of compound **2**, a proline-uncompetitive ligand class for *Hs*ProRS[22]. Although most analogs derived from this series showed moderate biochemical selectivity for *Hs*ProRS in the absence and presence of proline, the tested compounds were more selective for *Plasmodium* in cell-based assays (e.g., NCP26 $EC_{50,Dd2-2D4} = 67.4$ nM and NCP26 $EC_{50,HuH7} > 2.5$ μM).

Ultimately, cell-based selectivity is the more critical metric for drug development and increased cellular sensitivity of *Plasmodium* to ProRS inhibition has previously been noted for other ProRS inhibitors including halofuginone. While the higher sensitivity can in part be attributed to the increased sensitivity of rapidly proliferating cells to aaRS inhibition[25], and potentially favorable differential cellular

pharmacokinetics, much of the apparent disconnect between biochemical and cellular selectivity can be explained when considering the differential substrate affinities of the host and parasite ProRS paralogs. The physiological concentration of ATP in human cells (0.5 to 5 mM) and *P. falciparum* (2.5 mM) are comparable[43]. However, as we have shown, *Hs*ProRS binds ATP with much higher affinity than *Pf*cProRS does and is therefore predicted to exhibit much stronger competition with ATP-site directed ligands, such as NCP26[44]. The relevance of this effect has been well-documented for kinase inhibitors, where the disregard for differential $K_M$-values has resulted in misinterpretation of cellular activities[45,46].

In addition, we have validated *Pf*cProRS inhibition as the primary mode of action of NCP26 by a combination of in vitro resistance selection studies and whole genome sequence analysis. Although the evolution of resistance in response to drug treatment is never desirable, only very few antimalarials fail to induce in vitro resistance and recent guidelines have been developed to assess the risk for new candidates[36]. Compared to other drug-target combinations, NCP26 exhibited a very low propensity to induce resistance, requiring a large population size and long timeframe for resistance to arise (>50 generations)[36–38]. These findings are encouraging and demonstrate that the rapid resistance in response to *Pf*cProRS inhibition, as observed with halofuginone, is likely an inhibitor-class specific issue.

More recently, following our public disclosure of NCP26 and analogs, another group reported a related series of ProRS inhibitors, which also demonstrated dual-stage activity (liver and ABS)[47,48]. Although the authors did not provide direct biochemical evidence for *Pf*cProRS inhibition, they demonstrated oral efficacy of their lead compounds in a murine model of human malaria, which showed equal potency as NCP26 in ABS parasites.

Moreover, by formally linking halofuginone with adenosine or NCP26, we have been able to develop true triple-site aaRS inhibitors, i.e., ligands that meaningfully occupy all three substrate-binding pockets of the active site simultaneously. Such triple-site aaRS inhibitors have been postulated previously, but have remained elusive[39]. ThrRS ligands, such as borrelidin and analogs, which are triple-competitive and bind the amino acid and tRNA sites of ThrRS in addition to an auxiliary site, have been reported[39,49]. However, to the best of our knowledge, no triple-site aaRS inhibitors that target all substrate pockets have been reported to date[39]. Our co-crystal structure of MAT436 and *Pf*cProRS confirms the predicted binding mode. As expected, the triple-site compounds bound both ProRS paralogs with sub-nanomolar affinity in the absence of either substrate (proline or ATP). Their high biochemical affinity translated well into antiparasitic activity against both ABS and liver stage *Plasmodium* parasites, further validating the utility of our TR-FRET-based ligand displacement assay (Supplementary Fig. 16b). As before, we observed high selectivity for *Plasmodium* parasites over human cells in the cell-based assays. While we appreciate that the molecular mass of these triple-site inhibitors exceeds the canonical rule of five cut-off of 500 Da, consensus based on the analysis of approved drugs agrees that this limit has been overly stringent and suggest that the size of the hybrid ligands to be well within an acceptable range[50].

Our mechanistic analysis established a conclusive rationale for the selectivity increase of ProRS inhibitors that is observed when comparing biochemical and cellular activity data. However, future drug development efforts will nonetheless benefit from increased biochemical selectivity for *Pf*cProRS. While beyond the scope of this manuscript, we note that the TR-FRET tracers MAT379 and MAT425 exhibited >15-fold and >50-fold reduced affinity for *Hs*ProRS relative to *Pf*cProRS, respectively. This points to contributions of protein features adjacent to the active site. In fact, the region expected to be occupied by the FITC-functionalized linker represents one of the least conserved regions between *Pf*cProRS and *Hs*ProRS (Supplementary Fig. 1). Together, our data lay a strong foundation and provide a clear path

forward for the advanced development of ProRS inhibitors for malaria and potentially other human diseases with unmet medical needs.

## Methods

### Reagents and chemical synthesis

All reagents were purchased from Chem-Impex International Inc., Combi-Blocks Inc., Oakwood Chemical, Sigma-Aldrich, Fisher Scientific International Inc., VWR International, and BioSynth CarboSynth and were used without purification. Detailed synthetic procedures can be found in Supplementary Information. Stock solutions of inhibitors were prepared at 10 mM in molecular biology grade DMSO (Sigma-Aldrich). Halofuginone (**1**) was purchased from BioSynth CarboSynth and used without purification. Glyburide (**28**) was purchased from Combi-Blocks Inc. and used without further purification.

### Preliminary docking studies

While the inhibitor classes were all rationally designed, we conducted preliminary docking studies to guide the prioritization of which analogs to synthesize first. These studies were conducted in Spark™ v10.5.0, Forge™ v10.5.0, and Flare™ v4.0.2 (Cresset Biomolecular Discovery Ltd) per the manufacturer's instructions in their respective user guides. Ligands were docked against the ProRS structures reported here (PDB 6T7K, 7QB7, 7QC1, and 7QC2) and previously (for *Hs*ProRS, PDB: 5VAD, 4HVC, 4K86, 4K87, 4K88, and 5V58; for *Pf*cProRS, PDB 4Q15, 4NCX, 4YDQ, 4OLF, 5IFU, and 4WI1). Protein preparation was accomplished using default settings and the pharmacophore constraints were automatically generated and used without modification. Conformation hunts were done with VERY ACCURATE BUT SLOW setting modified to allow rotation about acyclic secondary amide bonds. Alignments were performed using both NORMAL (unbiased) and SUBSTRUCTURE (guided by ligands from crystal structures) settings. No model building was used to guide chemical synthesis.

### Protein constructs, expression, and purification

Genes encoding for *Hs*ProRS (residues 996–1512)[22], UniProt accession ID P07814) and *Pf*cProRS (residues 249–746, PF3D7_1213800)[24] were codon optimized for expression in *E. coli* and subcloned (GenScript Biotech Corporation, Piscataway, New Jersey) into a pFN29A His6-HaloTag T7 Flexi Vector (Promega), which contains an *N*-terminal His6-Tag-HaloTag (henceforth HT) followed by a linker sequence containing a TEV-cleavage site (5′-GAGCCAACCACTGAGGATCTGTACTTTCA-GAGCGATAACGCGATCGCC-3′). Expression plasmid sequences are included in the Source Data.

The HT-*Pf*cProRS and HT-*Hs*ProRS plasmids were independently transformed into SoluBL-21™ *E. coli* (Genlantis Inc. #C700200) and single colonies were picked from lysogeny broth (LB)-agar-ampicillin plate. SoluBL-21™ *E. coli* expressing either HT-*Pf*cProRS or HT-*Hs*ProRS were cultured at 225 rpm in lysogeny broth supplemented with 100 µg/mL ampicillin at 37 °C until OD$_{600}$ ~0.17, cooled to 15 °C, induced with 0.1 mg/mL IPTG (isopropyl β-D-thiogalactopyranoside), and cultured overnight at 15 °C. Cell pellets were collected via centrifugation for 20 min at 2800 × *g*, flash frozen with liquid nitrogen, and stored at −80 °C until lysis performed.

Bacterial cell pellets were quickly thawed in room temperature water and independently lysed on ice in B-PER Bacterial Protein Expression Reagent (Thermo Scientific #78243), pH 7.0 supplemented with 10 mM imidazole, 500 mM NaCl, 10 mM MgSO$_4$, 1 mM AEBSF, 1 mM dithiothreitol, 10% glycerol, 2 mM ATP, 25 U/mL Benzonase nuclease (Sigma-Aldrich #E1014), and 9000 U/mL Ready-Lyse lysozyme (VWR International #76081-780) until visually homogeneous and then for an additional 5 min. During lysis, samples were vortexed, sonicated, and pipetted up and down using a serological pipette. Cell lysates were clarified by centrifugation (2800 × *g* for 20 min followed by 21,000 × *g* for 10 min).

Clarified lysate was purified on HisTrap HP column (VWR #89501-388). All steps were performed at 1 mL/min and 4 °C. The column was pre-equilibrated with 10 mL water and 5 mL 10 mM imidazole in wash buffer (25 mM HEPES, pH 7.0, 500 mM NaCl, 1 mM DTT, and 10% glycerol). After clarified lysate was loaded, successively eluted with 5 mL of 20 mM imidazole in wash buffer, 5 mL of 40 mM imidazole in wash buffer, 5 mL of 300 mM imidazole in wash buffer, and then 5 mL of 500 mM imidazole in wash buffer.

Protein purity was assessed by sodium dodecyl sulfate polyacrylamide gel electrophoresis (SDS-PAGE) with 1.0-mm NuPAGE 4–12% Bis-Tris protein gels in NuPAGE MOPS running buffer at 120 V. The HaloTag of HT-ProRS was labeled prior to sample preparation with 100 µM TAMRA-Halo (**55**) for 15 min at room temperature[32]. Gels were analyzed using an Amersham Typhoon FLA 9500 fluorescence gel scanner (Cytiva Life Sciences; version 1.0.0.7; Cy3 excitation/emission) followed by Coomassie staining with SimplyBlue™ SafeStain (Thermo Fisher #LC6060). Fluorescent gel images were analyzed and quantified in ImageJ (Version 1.440, National Institutes of Health).

Protein concentration was measured by NanoDrop 1000 (Thermo Fisher Scientific, version 3.8.1) per the manufacturer's instructions (HT-*Pf*cProRS: molecular mass = 86.1 kDa and extinction coefficient = 155,000 M$^{-1}$ cm$^{-1}$; HT-*Hs*ProRS: molecular mass = 94.7 kDa and extinction coefficient = 150,000 M$^{-1}$ cm$^{-1}$).

Desired fractions based up were buffer exchanged into 25 mM HEPES, pH 7.0, 100 mM NaCl, 1 mM dithiothreitol, and 5% glycerol using PD-10 columns (Cytiva #17-0851-01) per the manufacturer's instructions.

For long-term storage, ProRS protein stocks were aliquoted following addition of glycerol to 20%, flash frozen in liquid nitrogen, and stored at −80 °C.

For crystallization studies, HT-*Pf*cProRS was expressed in a phage-resistant derivative of *Escherichia coli* strain BL21(DE3) carrying the pRARE2 plasmid for rare codon expression. Cells were grown at 37 °C in Terrific Broth supplemented with 100 µg/mL ampicillin until the culture reached an OD$_{600}$ of 2.0. The temperature was then decreased to 18 °C and protein expression induced with 0.5 mM IPTG (isopropyl β-D-thiogalactopyranoside) overnight. Cells were collected by centrifugation and resuspended in 50 mM HEPES, pH 7.5, 500 mM NaCl, 10 mM Imidazole, 5% glycerol, 0.5 mM TCEP, a protease inhibitor cocktail (Sigma), lysozyme, and benzonase, and lysed by sonication. The cell lysate was clarified by centrifugation and the proteins purified by nickel-affinity chromatography (Cytiva) using a stepwise gradient of imidazole. The His6-Tag-HaloTag fusion (HT) was removed by incubating with TEV protease at 4 °C overnight and this was followed by size exclusion chromatography (Superdex 200, Cytiva) in 20 mM MES, pH 6.0, 250 mM NaCl, 5% glycerol, and 0.5 mM TCEP. The TEV protease, cleaved byproducts containing histidine tag, and unreacted HT-*Pf*cProRS were removed by nickel-affinity chromatography and concentrated using an Amicon centrifugal filtration unit. The mass of purified protein was verified by electrospray ionization time of flight mass spectrometry (ESI-TOF-TOF: Agilent LC/MSD).

### CoraFluor-1-Halo labeling of HT fusion proteins

A freshly thawed solution of HT-ProRS in storage buffer (25 mM HEPES, pH 7.0, 100 mM NaCl, 1 mM dithiothreitol, and 20% glycerol) was incubated with 5 molar equivalents CoraFluor-1-Halo[32] overnight at 4 °C. Unreacted CoraFluor-1-Halo was removed by buffer exchanging with 7k MWCO Zeba spin desalting columns (Thermo Fisher Scientific #89883) into 25 mM HEPES, pH 7.0, 100 mM NaCl, 1 mM dithiothreitol, and 5% glycerol according to manufacturer's protocol.

Protein concentration following CoraFluor-1-labeling was semi-quantitatively measured by NanoDrop and corrected for absorbance by CoraFluor-1-Halo using Eq. (1) where the correction factor is the ratio of $A_{280}/A_{340}$ = 0.1571 for CoraFluor-1-Halo and $\varepsilon$ = extinction coefficient (for HT-*Pf*cProRS, $\varepsilon$ = 155,000 M$^{-1}$ cm$^{-1}$; for HT-*Hs*ProRS,

$\varepsilon = 150{,}000\,\mathrm{M^{-1}\,cm^{-1}}$[32]:

$$\text{Protein concentration in } \mu\mathrm{M} = \frac{A_{280} - (A_{340} * \text{correction factor})}{\varepsilon * \text{path length}} * 10^6 \tag{1}$$

The concentration of active HT-ProRS concentration following CoraFluor-1-Halo-labeling was quantitively measured by active-site titration of CoraFluor-labeled HT-ProRS (200 nM by nanodrop) with ProSA (**25**) in the presence of 250 nM MA379 (2.5x $K_D$ for HT-*Pf*cProRS and 0.15x $K_D$ for HT-*Hs*ProRS) to determine the IC$_{50}$, calculating the apparent $K_D$ using the Cheng−Prusoff equation, and doubling the apparent $K_D$ value (Eq. (2))[44]. This allowed for accurate $K_D$ determination for inhibitors suffering from ligand depletion (Eq. (3)) in the TR-FRET assay[51]:

$$[\text{active HT-ProRS}] = 2 \times \frac{\mathrm{IC}_{50}}{1 + \frac{[\mathrm{MAT379}]}{K_{\mathrm{D,MAT379}}}} \tag{2}$$

$$\mathrm{IC}_{50,\text{Ligand Depletion}} < 10 \times [\text{active HT-ProRS}] \tag{3}$$

For long-term storage at −80 °C, glycerol was added to 20% and samples were flash-frozen with liquid nitrogen.

## Time-resolved Förster resonance energy transfer (TR-FRET) measurements
Experiments were performed in white, 384-well microtiter plates (Corning 3572 or Greiner 781207). TR-FRET measurements were acquired on a Tecan SPARK plate reader with SPARKCONTROL software version V2.1 (Tecan Group Ltd.), with the following settings adapted from our previous TR-FRET experiments on other proteins[32]: 340/50 nm excitation, 490/10 nm (Tb) and 520/10 nm (FITC) emission, 100 μs delay, 400 μs integration. The 490/10 and 520/10 emission channels were acquired with a dichroic 510 mirror, using independently optimized detector gain settings. The TR-FRET ratio was taken as the 520/490 nm intensity ratio on a per-well basis.

## Determination of equilibrium dissociation constant ($K_D$) of tracers MAT379 and MAT425 toward ProRS paralogs by TR-FRET
A stock solution containing CoraFluor-1-Halo-labeled HT-*Pf*cProRS or CoraFluor-1-Halo-labeled HT-*Hs*ProRS at the concentration specified for each experiment was prepared in assay buffer (50 mM Tris, pH 7.5, 20 mM KCl, 10 mM MgCl$_2$, 0.05% Tween-20, 1 mM dithiothreitol, and 0.5 mg/mL BSA). A Multidrop Combi Reagent Dispenser (Thermo Fisher Scientific) was used to dispense 40 μL protein solution into wells of a white, 384-well plate (Corning 3572). Tracer MAT379 (**24**) or MAT425 was dispensed in dose-response in sextuplicate using a D300 digital dispenser (Hewlett Packard). Half the wells received 10 μM ProSA for background correction. Plates were mixed on an Ika MTS 2/4 Digital Microtiter Shaker at 750 rpm for 2 min, centrifuged at 1000 × $g$ at 25 °C for 1 min, and allowed to equilibrate at room temperature for 2 h before TR-FRET measurements were taken. Specific signal was determined by subtracting raw values from wells containing 10 μM ProSA (**25**). PRISM (V. 8.4.3−9.1.0, GraphPad) was used to perform non-linear regression analysis (one-site-specific binding), plot dose-response curves, and calculate $K_D$ values.

## Determination of Pro and ATP equilibrium dissociation constants ($K_D$ values) for individual, recombinant ProRS paralogs via TR-FRET-based ligand displacement assay
A stock solution containing CoraFluor-1-Halo-labeled HT-*Pf*cProRS (5 nM) or CoraFluor-1-Halo-labeled HT-*Hs*ProRS (1.5 nM) and tracer MAT379 (50 nM for HT-*Pf*cProRS and 1000 nM for HT-*Hs*ProRS) was prepared in assay buffer (50 mM Tris, pH 7.5, 20 mM KCl, 10 mM MgCl$_2$, 0.05% Tween-20, 1 mM dithiothreitol, and 0.5 mg/mL BSA). Prepared serial dilutions (octuplicate, 18-point, 1:2) of proline ($c_{\max} = 50$ mM) and ATP ($c_{\max} = 100$ mM) and dispensed 20 μL per well into flat, white, 384-well plates (Corning 3572). A Multidrop Combi Reagent Dispenser (Thermo Fisher Scientific) was used to dispense 20 μL of 2× protein solution into each well. Using a D300 digital dispenser (Hewlett Packard), dispensed 10 μM ProSA into half the wells of each dose of substrate for background correction. Plates were mixed on an Ika MTS 2/4 Digital Microtiter Shaker at 750 rpm for 2 min, centrifuged at 1000 × $g$ at 25 °C for 1 min, and allowed to equilibrate for 2 h at room temperature before TR-FRET measurements were taken.

PRISM (V. 8.4.3−9.1.0, GraphPad) was used to perform non-linear regression analysis (log(inhibitor) vs. response−Variable slope (four parameters)), plot dose-response curves, and calculate IC$_{50}$ values. The Cheng−Prusoff equation was used to convert IC$_{50}$ to $K_D$ values (Eq. (4)):

$$K_D = \frac{\mathrm{IC}_{50}}{1 + \frac{[\mathrm{MAT379}]}{K_{\mathrm{D,MAT379}}}} \tag{4}$$

## Determination of ProRS affinity and substrate-binding mode by time-resolved Förster resonance energy transfer assay
We utilized our TR-FRET assay to determine the affinity (equilibrium dissociation constants, $K_D$ values) and binding mode (competitive or noncompetitive vs. uncompetitive) with respect to substrates (proline and ATP) of our test compounds. Note that the substrate concentrations used here (0 μM proline + 0 μM ATP; 100 μM proline + 0 μM ATP; or 0 μM proline + 500 μM ATP) were chosen to identify uncompetitive inhibitors without substantial competition to the tracer MAT379 which is competitive with both ATP and proline. However, the ATP and proline concentrations used are not substantially above the substrates $K_D$ values to facilitate differentiation of substrate-noncompetitive and substrate-competitive inhibitors because these substrate concentrations would compete with our tracer and because we explicitly sought to develop proline-uncompetitive ProRS inhibitors to circumvent or overcome halofuginone-resistance mechanisms.

A stock solution containing CoraFluor-1-Halo-labeled HT-*Pf*cProRS or CoraFluor-1-Halo-labeled HT-*Hs*ProRS at the concentration specified for each experiment and 250 nM tracer MAT379 (**24**) was prepared in assay buffer (50 mM Tris, pH 7.5, 20 mM KCl, 10 mM MgCl$_2$, 0.05% Tween-20, 1 mM dithiothreitol, and 0.5 mg/mL BSA), and, where indicated, supplemented with 100 μM proline or 500 μM ATP. A Multidrop Combi Reagent Dispenser (Thermo Fisher Scientific) was used to dispense protein solution (30 or 40 μL) into each well of a flat, white, 384-well plate (Corning 3572 or Greiner 781207). Test compounds were dispensed in duplicate, triplicate, or sextuplicate dose-response format using a D300 digital dispenser (Hewlett Packard). Each plate included blank wells (no-inhibitor negative control for assay ceiling) and wells receiving 10 μM ProSA (**25**, positive control for assay floor) for Z-factor determination and a dose-response of NCP26 (**3**) as a standard. Plates were mixed on an Ika MTS 2/4 Digital Microtiter Shaker at 750 rpm for 2 min, centrifuged at 1000 × $g$ at 25 °C for 1 min, and allowed to equilibrate for 2 h at room temperature before TR-FRET measurements were taken.

Z-factors were calculated in Excel using 10 μM ProSA wells and negative control wells[34].

PRISM (V. 8.4.3−9.1.0, GraphPad) was used to perform non-linear regression (log(inhibitor) vs. response−Variable slope (four parameters)), plot dose-response curves, and calculate IC$_{50}$ values. The ligand-depletion corrected Cheng−Prusoff equation was used to convert IC$_{50}$ to $K_D$ values (Eq. (5)). $K_{\mathrm{D\_app,MAT379}}$ is defined as MAT379's $K_D$ corrected for the concentration of proline or ATP, if any, using the Cheng−Prusoff equation. Note that for Eq. (5), the [active HT-ProRS] was the active ProRS concentration determined by titration with ProSA

(see above)[52]:

$$K_D = \frac{IC_{50} - 0.5 \times [\text{active HT-ProRS}]}{1 + \frac{[\text{MAT379}]}{K_{D\_app,\text{MAT379}}}} \quad (5)$$

The inhibition mode for each test compound with respect to ATP or proline was determined by comparing the $K_D$ values measured in the presence and absence of each substrate.

As a reminder, Eq. (5) is only valid when the active HT-ProRS concentration is >~2 × $K_D$[51]. All values reported in the text or tables are not from ProRS-titrating conditions, but in some plots, compounds are titrating and these are clearly indicated in the figure legend (Fig. 2d–f, i, j and Supplementary Figs. 6, 11–13, 15). The anti-His6 antibody format (see below) was utilized to enable accurate determination of ProSA's affinity ($K_D$ value) and these data are shown for ProSA in Fig. 2g, k, Table 1, and Source Data–Inhibitor Profiling.

### Time-resolved Förster resonance energy transfer (TR-FRET) inhibition mode determination–anti-His6 antibody format

This assay was generally conducted in the same manner as the CoraFluor-1-Halo format with minor differences. All assays were conducted in sextuplicate dose-response with CoraFluor-1-Halo-labeled HT-ProRS whose concentration was determined by titration with ProSA. Each well was supplemented with 1 nM CoraFluor-1-Pfp-labeled anti-His6 antibody before the 2 h incubation.

The commercially available anti-His6 antibody (Abcam ab18184) was labeled as described previously[32]. In short, the antibody was buffer exchanged with 7k MWCO Zeba spin desalting columns (Thermo Fisher Scientific #89883) into 100 mM NaHCO3, pH 8.5, 0.05%v/v Tween-20 according to the manufacturers protocol. A 2.5 mM stock of CoraFluor-1-Pfp in dimethylacetamide (DMAc) was added to a final of 15 molar equivalents (eq) of CoraFluor-1-Pfp per antibody. Samples were mixed by briefly vortexing, pulse centrifuged, and incubated for 1 h at room temperature. Unreacted CoraFluor-1-Halo was removed by buffer exchanging with 7k MWCO Zeba spin desalting columns (Thermo Fisher Scientific #89883) into 25 mM HEPES, pH 7.0, 100 mM NaCl, 1 mM dithiothreitol, and 5% glycerol according to manufacturer's protocol.

After labeling, antibody samples were analyzed by Nanodrop as described above for HT-ProRS constructs. The following extinction coefficients were used to calculate antibody concentration and degree-of-labeling: Antibody $E_{280} = 210,000\ \text{M}^{-1}\text{cm}^{-1}$, CoraFluor-1-Pfp $E_{340} = 22,000\ \text{M}^{-1}\text{cm}^{-1}$. Antibody conjugates were diluted with 50% glycerol, flash-frozen in liquid nitrogen, and stored at −80 °C.

### TR-FRET binding kinetics

Dissociation rates ($k_{off}$) of the TR-FRET tracer MAT379 (24) and HT-ProRS homologs were measured by rapid dilution ($n = 23$) in white, 384-well plates. An equilibrated solution of 5 µL 100 nM CoraFluor-1-Halo-labeled HT-ProRS, ~EC80 MAT379 (560 nM MAT379 for HT-PfcProRS and 7 µM MAT379 for HT-HsProRS), and either 10 µM ProSA (~20,000 × $K_D$) or DMSO vehicle in assay buffer (50 mM Tris, pH 7.5, 20 mM KCl, 10 mM MgCl2, 0.05% Tween-20, 1 mM dithiothreitol, and 0.5 mg/mL BSA) was diluted 10-fold into 45 µL assay buffer using a multichannel pipette, and briefly mixed by pipetting up and down three times. TR-FRET measurements were acquired in kinetic mode (1 read every ~45 s) for at least 10 min. Excel was used to subtract the background signal (10 µM ProSA wells) from the DMSO vehicle wells. PRISM (V. 8.4.3–9.1.0, GraphPad) was used to perform non-linear regression (Dissociation–One Phase exponential decay), plot 520/490 nm TR-FRET ratio vs. time, and calculate $k_{off}$ values.

We attempted to measure the association rates ($k_{on,obs}$) using a similar method (described below), but they were too fast to measure (fully equilibrated by first time point) so we instead calculated the association rates ($k_{on,calc}$) using the measured dissociation rates ($k_{off}$)

and measured equilibrium dissociation constants ($K_D$) in Eq. (6):

$$k_{on,calc} = k_{off} / K_D \quad (6)$$

Our attempts to measure association rates ($k_{on,obs}$) of the TR-FRET tracer MAT379 and HT-ProRS homologs were similarly performed in dilution format ($n = 23$) in white, 384-well plates. An equilibrated solution of 5 µL 100 nM CoraFluor-1-Halo-labeled HT-ProRS and either 10 µM ProSA or DMSO vehicle in assay buffer was diluted 10-fold into 45 µL MAT379 (500 nM MAT379 for HT-PfcProRS and 5 µM MAT379 for HT-HsProRS) in assay buffer using a multichannel pipette, and briefly mixed by pipetting up and down three times. TR-FRET measurements were acquired in kinetic mode (1 read every ~45 s) for at least 10 min. Excel was used to subtract the background signal (10 µM ProSA wells) from the DMSO vehicle wells. PRISM (V. 8.4.3–9.1.0, GraphPad) was used to perform non-linear regression (Association kinetics–One Conc. of hot), plot 520/490 nm TR-FRET ratio vs. time, and calculate $k_{on,obs}$ values. However, as noted above, this failed to provide meaningful $k_{on,obs}$ values because the samples were fully equilibrated by the first scan, and thus too fast to measure.

### Crystallization, data collection and structure determination

PfcProRS was co-crystallized with NCP26 (3), MAT334 (29), MAT345 (30), and MAT436 (34) at 20 °C using the sitting drop vapor diffusion method.

For crystals of PfcProRS in complex with NCP26 and proline (PDB: 6T7K), 2 mM NCP26 was added to 39 mg/mL PfcProRS together with 5 mM L-Proline, and crystals were obtained in a drop containing 75 nL of protein-compound mixture and 75 nL precipitant composed of 0.1 M HEPES, pH 7.5, and 20% PEG 10,000.

For crystals of PfcProRS in complex with MAT334 and proline (PDB: 7QC2), MAT334 was added to PfcProRS (3 mg/mL) at a concentration of 0.5 mM, and the protein-compound mixture incubated 30 min on ice before it was concentrated to 28.5 mg/mL. Crystals of PfcProRS in complex with MAT334 and proline were obtained in a drop containing 75 nL of protein-compound mixture and 75 nL precipitant composed of 0.2 M L-Proline, 10% PEG3350, and 0.1 M HEPES, pH 7.5.

Crystals of PfcProRS in complex with MAT345 and proline (PDB: 7QB7) were obtained in a drop containing 75 nL of a protein-compound mixture with 1 mM of MAT345, 5 mM L-proline, and 22 mg/mL PfcProRS, and 75 nL precipitant compost of 25% PEG3350 and 0.1 M BIS-TRIS, pH 6.5.

For crystals of PfcProRS in complex with MAT436 (PDB: 7QC1), 1 mM MAT436 was added to 22.4 mg/mL PfcProRS, and crystals obtained in a drop containing 100 nL of the protein compound mixture and 50 nL of 1.5 M malic acid.

The crystals were cryo-protected in precipitant solution supplemented with 25–30% ethylene glycol and then flash cooled in liquid nitrogen. Data were collected on beamlines I03 and I04 at the Diamond Light Source UK, and the dataset processed, scaled, and merged at the Diamond Light Source using Xia2[53]. Electron density maps were obtained by molecular replacement using PHASER with previously determined structures of PfcProRS as a search model.

The complex structure of PfcProRS with NCP26 (PDB 6T7K) was solved to 1.79 Å resolution using PDB 4Q15 as a search model. The complex structure of PfcProRS with MAT334 was solved to 2.28 Å resolution (PDB 7QC2), MAT345 to 1.92 Å (PDB 7QB7), and MAT436 to 2.51 Å resolution (PDB 7QC1) using PDB 6T7K as search model. The structures were refined in an iterative process using PHENIX (1.19.2-4158) with electron density map inspections and model improvement in WinCOOT (0.9.4.1) and terminated when there were no substantial changes in the $R_{work}$ and $R_{free}$ values and inspection of the electron density map suggested that no further corrections or additions were justified[54,55]. Structural analysis and figures were performed with PyMOL (http://www.pymol.org).

 

Crystallographic data and refinement statistics are available in Supplementary Table 1.

## *P. falciparum* cell lines and culture conditions

Parasites were maintained under standard culture conditions as described previously[56,57]. The *P. falciparum* Dd2-2D4 clone was derived from Malaria Research and Reagent Resource Repository line MRA-156 (BEI Resources)[57]. The *P. falciparum* halofuginone-induced (elevated proline homeostasis) and HFGR-I (elevated proline homeostasis and *Pf*cProRS[L482H]) were previously reported[8,17,27]. The *P. falciparum* Dd2 cell lines used in this study (Dd2-2D4, halofuginone-induced, and HFGR-I) have previously undergone whole genome sequencing in our labs. They were also routinely assayed and EC$_{50}$ values determined using the ABS growth assay (see below) with a panel of standard antimalarial compounds, including halofuginone which has differential activity for the three Dd2 lines used in these experiments. Data were cross-referenced for consistency with literature reported values and data previously acquired in the lab.

The red blood cells used for culturing *P. falciparum* parasites were isolated from whole blood purchased from Interstate Blood Bank and was tested for blood borne pathogens including human immunodeficiency virus, hepatitis C virus, and hepatitis B virus.

The blood and *P. falciparum* Dd2 cell lines used for this study were not routinely tested for mycoplasma.

## *P. falciparum* asexual blood stage growth assay

This assay was performed as previously described[8]. In short, *P. falciparum* erythrocytic-stage parasites at 1% parasitemia and 1% hematocrit in RPMI + 0.5% Albumax were seeded at 40 μL/well in 384-well plates with test compounds in triplicate, dose-response format with 10 μM dihydroartemisinin as a kill-control and blank (no compound) wells as a growth-control. DMSO concentration did not exceed 1% (v/v). After 72 h, growth was quantified by measuring fluorescence (Molecular Devices SpectraMax iD5) following SYBR Green 1 staining. Data were analyzed in Excel and plotted in PRISM (V. 8.4.3–9.1.0, GraphPad).

## *P. falciparum* asexual blood stage short-term resistance susceptibility assay

Using the robust procedure previously used to generate halofuginone-induced parasites (halofuginone-tolerant with elevated proline homeostasis), we unsuccessfully attempted to generate NCP26-tolerant/resistant parasites[8,17,27]. In short, three independent flasks of *P. falciparum* Dd2-2D4 parasites (~3 × 10$^8$ parasites per independent selection flask) were treated with 4x EC$_{50}$ NCP26 until no parasites were detected by Giemsa staining microscopy. Following recrudescence, sensitivity to NCP26 and halofuginone was assayed using the ABS growth assay.

## NCP26 resistance selection

Three independent selections for NCP26-resistant mutants of *P. falciparum* Dd2-2D4 parasites were conducted in vitro as previously reported[8,27]. In short, parasites were treated with 4x EC$_{50}$ NCP26 until no parasites were detected by giemsa staining microscopy. Following recrudescence, the ABS growth assay was used to determine sensitivity to NCP26 and control compounds including ProRS inhibitors halofuginone (**1**), halofuginol (**26**), and ProSA (**25**); threonyl-tRNA synthetase (ThrRS) inhibitor borrelidin; and dihydroartemisinin (DHA). This cycle was repeated for ~50 generations (~100 days), corresponding to 5–6 cycles of drug pressure. We initially began the selections with ~3 × 10$^8$ parasites per flask (i.e., per independent selection), but did not observe any resistance after 2 cycles of drug pressure (38 days; ~19 generations) so we expanded the selection cultures to ~1 × 10$^9$ parasites per flask and maintained this for the remainder of the selection.

## Subcloning

Clonal parasites were isolated from each selection flask by limiting dilution of ring stage parasites in 96-well plates to an average of 0.8 and 0.2 parasites per well. Following recrudescence, these clonal parasites were assayed in the ABS blood stage viability assay to ensure no phenotypic differences from the corresponding bulk population (all isolated clones had EC$_{50}$ values for all inhibitors tested within two-fold of corresponding bulk population).

## Library preparation and whole genome sequencing

Infected RBCs were washed with 0.05% saponin and genomic DNA was isolated from the parasites using a DNeasy Blood and Tissue Kit (Qiagen) according to the standard protocols. Sequencing libraries were prepared with the Nextera XT kit (Cat. No FC-131-1024, Illumina) via the standard dual index protocol and sequenced on the Illumina NovaSeq 6000 S4 flow cell to generate paired-end reads 100 bp in length. Sequence data is available under BioProject Accession number: PRJNA811614 in the NCBI Sequence Read Archive. Reads were aligned to the *P. falciparum* 3D7 reference genome (PlasmoDB v13.0) using the previously described pipeline[38]. A total of eight samples were sequenced to an average whole genome coverage of 157×, with an average of 89% of reads mapping to the reference genome (Source Data−Sequencing Statistics). Following alignment, SNVs and INDELs were called using GATK HaplotypeCaller and filtered according to GATK's best practice recommendations[58]. Variants were annotated using a custom SnpEff database and further filtered by comparing those from resistant clones to the parent clone, such that only a mutation present in the resistant clone but not the sensitive parent clone would be retained. CNVs were identified by differential Log2 copy ratio as described in the GATK 4 workflow. Briefly, read counts were collected across genic intervals for each sample. Copy ratios were calculated after denoising read counts against a strain-matched Panel of Normals composed of non-drug-selected Dd2 parasite samples.

## PCR amplification and Sanger sequencing

Genomic DNA was isolated as described above (see Library preparation and whole genome sequencing). Sections of the *cPRS* gene were amplified by polymerase chain reaction (PCR) to validate the mutations observed by whole genome sequencing. Primers (single stranded DNA oligomers) were ordered from Integrated DNA Technologies Inc (see Source Data for sequences). Immediately prior to PCR reaction, combined 2.5 μL 5 μM forward primer + 5 μM reverse primer in 1x TE buffer (10 mM Tris, pH 7.5, 1 mM EDTA) or 2.5 μL 1x TE buffer (no primer control) with 10 μL 1 ng/μL isolated gDNA, mixed by briefly vortexing, and pulse spun. Added 12.5 μL 2x GoTaq G2 Colorless Mastermix, mixed by briefly vortexing, and pulse spun. PCR reactions were performed on thermocycler (Eppendorf AG 22331 Hamburg No. 5341) with the following method: 95 °C for 2 min; 30 cycles of 95 °C for 1 min, 55 °C for 1 min, and 73 °C for 1 min; and then 73 °C for 5 min.

PCR reactions were analyzed by 1% agarose gel electrophoresis and fluorescently imaged following ethidium bromide staining to ensure PCR reactions produced one product. DNA was purified from PCR reactions using Zymo DNA Clean and Concentrator-5 Kit (Zymo Research #D4005). Purified DNA was submitted to Genewiz Inc for Sanger sequencing and results were aligned to the predicted and sequenced results from the Dd2-2D4 parent line using Benchling.

Sanger sequencing data from Genewiz are provided in the Source Data file. Note that for the *Pf*cProRS[F405L] mutation (S3), direct analysis of the raw fluorescence intensity data was required as Sanger sequencing samples were called as wild-type by Genewiz despite having ~50% as much signal for the *Pf*cProRS[F405L] mutant allele. This is consistent with whole genome sequencing data.

## *P. berghei* liver stage and HuH7 host hepatocyte growth assay

HuH7 cells (Sigma, Cat#: 01042712, Lot:18H009) were cultured in DMEM + L-Glutamine (Gibco) supplemented with 10% (v/v) heat-inactivated FBS (Sigma) and 1% (v/v) antibiotic/antimycotic (Sigma). Hepatocyte cultures were maintained in a standard tissue culture incubator at 37 °C. *Anopheles* mosquitoes infected with luciferase-expressing *P. berghei* ANKA sporozoites were obtained from the Sporocore at the University of Georgia. Liver stage *P. berghei* assays were completed as previously described[59]. Briefly, 4000 HuH7 cells were seeded into 384-well plates (Corning) 1 day prior to infection. Compounds (0–50 μM) were added in triplicate to wells before infection with 4000 *P. berghei* sporozoites. At ~44 h post infection (hpi), HuH7 cell viability and *P. berghei* parasite load was assessed using CellTiter-Fluor (Promega) and Bright-Glo (Promega), respectively, using an Envision plate reader (Perkin Elmer, 1.13.3009.1401). Relative fluorescence and luminescence signal intensities were normalized to the negative control, 1% DMSO. $EC_{50}$ values were determined using PRISM (V. 8.4.3–9.1.0, GraphPad) through fitting data to a dose-response curve.

HuH7 cells tested negative for mycoplasma within 2 months before and after experiments.

## Determination of correlation between assays

The Spearman's rank correlation coefficients ($r_s$) were calculated using PRISM (V. 8.4.3–9.1.0, GraphPad) using only pyrazinamide compounds (i.e., excluding ProSA, D-ProSA, halofuginone, halofuginol, and glyburide). The TR-FRET $pK_D$ values used were from the highest affinity conditions (i.e., data from absence of substrates for ATP- and proline-competitive inhibitors, 100 μM Pro for proline-uncompetitive inhibitors, and 500 μM ATP for ATP-uncompetitive inhibitors). The $pEC_{50}$ values used were for ABS Dd2-2D4.

## General statistics

All confidence intervals were calculated using PRISM (V. 8.4.3–9.1.0, GraphPad) and are asymmetric. Additional statistical information specific to each experiment is provided in the corresponding protocol above.

## Reporting summary

Further information on research design is available in the Nature Research Reporting Summary linked to this article.

## Data availability

The authors declare that the main data supporting the findings of this study are available within the article, its Supplementary Information files, and the Source Data files. Extra data (raw plate reader data for biological experiments; NMR and LC/MS data files) are available from the corresponding author upon request. Images of the $^1H$ and $^{13}C$ NMR spectra are provided in the Supplementary Information. Expression plasmid sequences, normalized data for TR-FRET and cellular (*P. falciparum*, *P. berghei*, and HuH7) experiments, and Sanger sequencing data are available in the Source Data. The co-crystal structures have been deposited in the Protein Data Bank (PDB: 6T7K, 7QB7, 7QC2, 7QC1). Whole genome sequence analysis data is available in the NCBI Sequence Read Archive under BioProject number PRJNA811614. Primer sequences for PCR amplification and Sanger sequencing are available in the Source Data files. Source data are provided with this paper.

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

## Acknowledgements

We thank Dr. Paul Hinkson, Dr. Robert L. Summers, Dr. Rebecca E.K. Mandt, and Dr. Selina Bopp from the Wirth laboratory for experimental guidance and helpful feedback. This work was supported by NIH R01AI143723 (R.M. and D.F.W.), NIH R01AI152533 (M.R.L. and E.A.W.), 5F31AI129412 (L.F.), and the Bill & Melinda Gates Foundation (OPP1054480, E.A.W. and D.F.W.), LEAN program of the Leducq Foundation (U.O.), Arthritis Research UK 20522 (U.O.), Cancer Research UK A23900 (U.O.). N.C.P. was supported by a National Science Foundation Graduate Research Fellowship (DGE1745303). M.R.L. was supported in part by a Ruth L. Kirschstein Institutional National Research Award from the National Institute for General Medical Sciences (T32 GM008666). This publication includes data generated at the University of California, San Diego IGM Genomics Center utilizing an Illumina NovaSeq 6000 that was purchased with funding from a National Institutes of Health SIG grant (#S10 OD026929).

## Author contributions

Concept: R.M. Supervision: M.M.M., E.A.W., A.K.L., E.R.D., U.O., D.F.W., and R.M. Experiments: M.A.T., N.C.P., C.J., Kr.S., S.A.S., L.F., A.P., Ka.S., M.R.L., and S.M. Data analysis: M.A.T., C.J., L.F., A.P., M.R.L., M.W., M.M.M., E.A.W., A.K.L., E.R.D., U.O., D.F.W., and R.M. Manuscript writing: M.A.T and R.M. All authors read, revised, and approved the manuscript.

## Competing interests

R.M. is a scientific advisory board (SAB) member and equity holder of Regenacy Pharmaceuticals, ERX Pharmaceutics, and Frequency Therapeutics. M.A.T., N.C.P., D.F.W. and R.M. are inventors on patent applications related to this work. The remaining authors declare no competing interests.
