## [Peer Review File · Nature Communications]

Elucidating the path to Plasmodium prolyl-tRNA synthetase inhibitors that overcome halofuginone-resistanceReviewers' comments:

Reviewer #1 (Remarks to the Author):

Tye et al. tried to discover the new class inhibitors of human and Plasmodium prolyl-tRNA synthetases. As an effort to determine the exact KD value of the test compounds to ProRSs, the authors developed a novel biochemical assay platform. Although the authors showed the activities of the triple-site inhibitors for human and Plasmodium ProRSs, they did not show efficacy on halofuginone-resistant parasites. Thus, some evidences and suggestions should be additionally provided to clearly suggest the distinction and potential applicability of their approach and generated compounds.

Major comments:

- 1) The authors should clarify and clearly state the purpose of the study and evaluate the results based on the purpose. If the purpose was not achieved, the authors should well describe the reason based on the experimental results and analysis and suggest the possible direction of the next compound development.
- 2) Although the authors pointed the necessity of the development of next-generation antimalarials with multi-stage activity by targeting PfcProRS (the development of rapid resistance to halofuginone analogs along with poor biochemical selectivity for the parasite over host enzyme), the compounds analyzed in this work did not seem to show much improvement in the end. If the main purpose of this work was just the development of the new class of ProRS inhibitors, the authors should be clear about it.
- 3) The authors compared the amino acid sequences between HsProRS and PfcProRS. However, they gave little effort to develop compounds more specific for PfcProRS. At least, the authors could compare the similarities and differences of binding residues of ProRS between more and less selective compounds.
- 4) There are references explaining how halofuginone-resistant parasites gain resistance at molecular level. However, there are no efforts to utilize the information or rationally design the compounds based on these informations. At least, the authors should describe the experimental results based on the resistance mechanism and then further suggest the next direction to avoid the resistance mechanism.
- 5) The tracer must show the constant TR-FRET ratio irrespective of the existence of substrates. In addition, the assay conditions have to reflect the physiological conditions since the test compounds are either competitive or uncompetitive inhibitors that are affected by substrates concentrations. How much ATP and proline exist in Plasmodium and human cells? Are the assay conditions able to properly reflect the physiological conditions? What is the KD value of MAT379 to HsProRS in the presence of either proline or ATP? What is the KD value of MAT379 to PfcProRS in the presence of either 1 mM ATP or 500 μ M proline (or at the physiological range of the substrates concentrations). The authors suggested that the KD-values for ATP and proline are 892 μ M and 457 μ M, respectively. Is the physiological range of the substrates concentrations similar to those KD values?
- 6) The authors determined the KD values in three different conditions, 0 μ M Pro/ATP, 100 μ M Pro and 500 μ M ATP. Although this approach seems to be useful to study the MoA of the inhibitors, there are both proline and ATP in the actual working conditions of the inhibitors. The use of both proline and non-hydrolysable analog of ATP at the same time would derive much closer values to induce cytotoxic effect to the parasite/host and provide the direct evidence of parasite/host selectivity as the authors reasoned (Line 397, 398).
- 7) How does MAT379 compete with proline? MAT379 is derived from T-3767758 and even the pyrazinamide-proline hybrids (MAT334 and MAT345) were found to be not competitive to proline. Is there any theoretical evidence for that? The graph for the determination of KD values for ATP and proline of HsProRS should be provided in the Results section.

Minor comments:

1. MAT334 and MAT345 are not proline-competitive inhibitors. The complex structure and biochemical assay results support that MAT334 and MAT345 are not proline-competitive inhibitors. Although the authors described "unsuccessful attempts (Line 307)" in the end of the paragraph, they

should revise the manuscript to prevent the reader's misunderstanding by emphasizing that the compounds are designed to be proline-competitive and proved to be not proline-competitive through the series of experiments.

2. Line 330: Presentation of the chemical structure of compound 40 (iso-MT436) will help the readers to understand the paper.

3. Which compound is better for parasite/host selectivity? The direct comparison of the representative compounds (including control compounds like halofuginone and compound 2) of each class on parasite/host selectivity will provide useful information to the readers. The additional column in the Table 1 about parasite/host selectivity will be helpful.

4. Line 336: This sentence is ambiguous. The exact comparison statement will help the readers to understand the paper. (An overlay with the corresponding ternary complex of PfcProRS with halofuginone and AMP-PNP(PDB 4Q15) revealed a nearly perfect overlap with halofuginone)

5. Line 357: human hepatocyte liver cells \diamond human hepatocytes or human liver carcinoma cells

6. Line 651, Line 677: What is the meaning of "these conditions" in the following sentences?

(Line 651: Compounds marked with ** were titrating ProRS under these conditions.

Line 677: Compounds marked with * are titrating HsProRS under these conditions.)

7. Line 675: 2.5x KD \diamond 0.15x KD

8. The methods for determining the dissociation rates and association rates seem to be the same. What's the difference?

Reviewer #2 (Remarks to the Author):

Tye et al have prepared a well written manuscript that focusses on development of new inhibitors against malaria parasites. The experiment rationale and design are clear although the general ideas have been published earlier.

Some noteworthy results include:

1. Discovery of novel ATP mimetics that are proline uncompetitive based on pyrazinamide-based inhibitors that stem from previously published work from Takeda Pharmaceuticals where they discovered ATP-based inhibitors of PRS.

2. Discovery of a novel triple site inhibitor for PRSs that is essentially an ATP mimetic moiety linked to a halofuginone-quinazolinone moiety by a sulfamoylcarbamate group.

3. Parasite and host cellular inhibition tested against different cell-lines (Dd2-2D4, HFG-induced and HFGR-I Pf cell lines, Pb liver stage tissue culture and HuH7 host cell line) are described with these compounds.

4. A TR-FRET assay platform has been described but whether the reagents used are in public domain and/or accessible to all is unclear.

This work will further efforts for development of selective triple site inhibitors of PRS but as yet this has not been achieved, thus limiting the impact of this work.

Some major issues that need to be addressed:

1. For the KD experiment, why were these concentrations (100 μ M L-pro or 500 μ M ATP) chosen for TR-FRET? Is it just to have excessive natural substrate or any other reasoning exists? And if it is for saturation – why were these exact particular values chosen? Is there any correlation with the substrate K_m here? Considering these scaffolds are competitive with either of the natural substrates, the relative concentrations of the natural substrate and the enzyme might have differential effects on the measured KD.

2. Furthermore, KD should have been determined in another combination as well – in presence of both natural substrates (L-pro and ATP at ~2.5Km concentration ranges as indicated in ref 22 “How to measure and evaluate binding affinities”) and in presence of tRNAPro. Currently, the data points only suggest a very good association between the protein and drugs, however, this does not categorically prove biochemical inhibition.

3. Enzyme inhibition assays are missing and hence there is no biochemical evidence (despite the structural evidence) of the mode of action of the compounds used in this study. The compounds do bind (as per structures) to PRS but whether they actually inhibit the enzyme activity remains to be shown. The authors themselves realise that even a previous study (ref 20, 21, 26, 39) did not have these relevant data (Page 17 line 402-404 : “Although the authors did not provide direct biochemical evidence for PfcProRS inhibition, they demonstrated oral efficacy of their lead compounds in a murine model of human malaria, which showed equal potency as NCP26 in ABS parasites”. Biochemical inhibition data will be vital in going forward with these compounds and their derivatives as well as doing 3D structures of all possible hit compounds will be tedious.

In addition:

1. Lines 75-77: “Crystallographic data of the co-complexes with human and Plasmodium ProRS revealed that halofuginone binds the A76-tRNAPro 76 and proline-binding pockets of the active site (Fig. 1a), which are highly conserved between both homologs.” The appropriate ref for this is Jain et al who resolved the PfPRS-HFG complex (PDB ID: 4YDQ) (Structure. 2015 May 5;23(5):819-829. doi: 10.1016/j.str.2015.02.011. Epub 2015 Mar 26).

2. In supplementary Figure 1- the ribbon may be hidden for clarity. The surface transparency can be decreased to show the bound ligands clearly, particularly bound proline.

3. Line 695: Supplementary Table 1 was mentioned but this is missing in supplementary data.

4. Supplementary Figure 2. It would be better for the reader to connect the dots if a similar annotation (circle/square/triangle) is used for the same compounds in different panels. That way a consistency and continuity would remain.

5. In Supplementary Table 2:

a) Space group notation – screw axis notation should be in subscript.

b) “l / s (l)” should be replaced by “l / δl”.

c) The symbol “” should be removed in “51.68-`1.79”.

d) Decimal approximation may be needed for the average B-factor values.

6. Supp_Methods: Line 167: D should be subscript in KD, i.e. KD should be KD.

Reviewer #3 (Remarks to the Author):

This is an interesting paper that greatly advances inhibitor work for Plasmodium falciparum (malaria) Prolyl-tRNA-synthetase (ProRS). The paper merges: 1) assay development of a unique FRET assay that can be used across many AARS targets that solves many issues with previous assays; 2) moving from a good industry lead to leads that are more potent; 3) using structural biology and enzyme

assays to characterize the binding properties, including which of the 3 pockets (tRNA vs. Proline vs. ATP pockets); and finally, 4) using structure-guided drug optimization to make a unique binder of PfProRS that reaches into all 3 pockets. The paper is greatly aided by analysis of wild-type Pf parasites and parasites that have elevated proline levels and also a mutant PfProRS and parasites that express the mutant. It is clear that triple-pocket binding compounds are rendered ineffective by these mutant Pf strains. My only suggestion about improving the paper is that it took me 2-3 readings of the paper to get the basic points. There are a lot of details in the main body of the paper that detract from readability. The authors could move some of the details into supplement, so it doesn't break up the flow of the paper. In my opinion, this would improve readability.

RESPONSE TO REVIEWERS' COMMENTS

Reviewer #1 (Remarks to the Author):

Tye et al. tried to discover the new class inhibitors of human and Plasmodium prolyl-tRNA synthetases. As an effort to determine the exact KD value of the test compounds to ProRSs, the authors developed a novel biochemical assay platform. Although the authors showed the activities of the triple-site inhibitors for human and Plasmodium ProRSs, they did not show efficacy on halofuginone-resistant parasites. Thus, some evidences and suggestions should be additionally provided to clearly suggest the distinction and potential applicability of their approach and generated compounds.

While it is true that some of our inhibitors (e.g. triple-site inhibitor MAT436) were cross-resistant with halofuginone, others (e.g. NCP26) did not display reduced activity in halofuginone-resistant parasites (halofuginone-induced and HFGR-I). Thus, we have confirmed our hypothesis that the rapid halofuginone-tolerance observed following halofuginone treatment is not a universal problem for PfcProRS inhibitors, but rather is specific to proline-competitive PfcProRS inhibitors like halofuginone itself.

In our original manuscript we also reported that NCP26 does not induce short-term resistance in Plasmodium parasites (Line 140: “Moreover, unlike halofuginone, NCP26 did not induce rapid resistance in wild-type parasites that were subjected to intermittent drug pressure.”).

Since the submission of our original manuscript, we have conducted extended resistance selection studies and provided additional evidence that *P. falciparum* exhibits greatly reduced propensity to evolve resistance under NCP26 drug pressure compared to halofuginone (> 50 generations with 10^9 parasites versus < 5 generations with < 10^8 parasites, respectively). Furthermore, we have performed whole genome sequence analysis on several parasite clones from the independent selection experiments, which identified single amino acid changes, PfcProRS^{F405L} or PfcProRS^{T512S}, in the NCP26 binding site (ATP pocket) as resistance mutations.

These data, which have been included in the revised manuscript, provide additional strong support for our claim that NCP26 is acting through PfcProRS as the principle functional target *in vitro*, and that ATP-competitive, proline-uncompetitive PfcProRS inhibitors are a promising strategy for developing anti-malarial chemotherapeutics.

Major comments:

1) The authors should clarify and clearly state the purpose of the study and evaluate the results based on the purpose. If the purpose was not achieved, the authors should well describe the reason based on the experimental results and analysis and suggest the possible direction of the next compound development.

The purpose of this study was to establish a path forward for the development of PfcProRS inhibitors. Current progress has been hampered by the lack of a robust, flexible, and quantitative biochemical assay platform, and programs have been further discouraged by the rapid development of resistance to halofuginone-based inhibitors. Our stated goals were:

- **Development of an assay platform that would overcome limitations of existing technologies and enable facile, quantitative characterization of ProRS inhibitors.**
- **Development of new inhibitor classes starting from existing leads and to evaluate:**
 - **The ability to overcome halofuginone-resistance**
 - **The propensity to evolve resistance**
 - **An approach that will enable biochemical selectivity over the human paralog (in addition to cellular selectivity)**
 - **New inhibitor concepts, including speculative tri-site inhibitors, which have been proposed as promising ligands for this enzyme class yet have remained elusive to develop**

All stated objectives have been accomplished. Together this data will support and guide future development against this target class.

2) Although the authors pointed the necessity of the development of next-generation antimalarials with multi-stage activity by targeting PfcProRS (the development of rapid resistance to halofuginone analogs along with poor biochemical selectivity for the parasite over host enzyme), the compounds analyzed in this work did not seem to show much improvement in the end. If the main purpose of this work was just the development of the new class of ProRS inhibitors, the authors should be clear about it.

As we discuss in our manuscript, the fundamental issue with halofuginone analogs is not primarily biochemical selectivity (in the absence of elevated concentrations of competing substrates) as they exhibit sufficient selectivity in a cellular context. Rather, the issue pertains to the rapid development of resistance (as we previously reported in references 8, 16, and 27). Overcoming this challenge was one of our stated goals. Thus, we took two approaches to address this problem:

- 1. To develop inhibitors that are not cross-resistance with halofuginone by either mechanism (elevated proline homeostasis or *PfcProRS*^{L482H} mutation in proline pocket) and exhibit low propensity to induce other modes of resistance.**
- 2. To identify new leads with improved potency and biochemical selectivity to further expand the therapeutic window.**

We successfully identified several compounds with improved potency, including NCP26, that are not cross-resistant with halofuginone. As noted above, in our revised manuscript we expand on the short-term selection studies and conducted extended large scale resistance selections that demonstrate NCP26's greatly reduced risk for resistance evolution. While it will be desirable to further improve biochemical selectivity, and our results provide a rational path towards this goal, we demonstrate that our inhibitors exhibit high cellular selectivity, which is the more important metric for drug development.

As we discuss in our manuscript and in our detailed response to Major Comments 5 and 6, the apparent contradiction between cellular and biochemical selectivity is likely not only the result of decreased sensitivity to aaRS inhibition in slow-proliferating cells (and potentially differential cellular pharmacokinetics), but can be readily rationalized by considering the presence of competing substrates and their differential affinity for the host and parasite enzymes. NCP26 is ATP-competitive

and binds both ProRS paralogs with comparable affinity to one another in the absence and presence of proline. However, *HsProRS* has much higher affinity for ATP, and given the similar cellular ATP concentrations in host and parasite, will be less sensitive to NCP26 in cellular context.

3) The authors compared the amino acid sequences between *HsProRS* and *PfcProRS*. However, they gave little effort to develop compounds more specific for *PfcProRS*. At least, the authors could compare the similarities and differences of binding residues of ProRS between more and less selective compounds.

One of the stated main objectives was the identification of inhibitors that are refractory to halofuginone-resistance mechanisms (see above). As noted in our original manuscript, the active site is highly conserved between the parasite and host paralogs. The ATP-binding pocket is the only portion of the active site that is not fully conserved, but it still displays very high homology (PMID: 23263184 and 25047712), which is why we focused on ATP-competitive ProRS inhibitors. Further, we demonstrated that MAT379 was >15-fold selective for *PfcProRS* over *HsProRS in vitro*, and in SI Figure 1, we highlighted the cluster of non-conserved residues near the ProRS active site pore which we hypothesize are responsible for MAT379's selectivity. In our revised manuscript, we also included new data surrounding MAT425, an analog of our TR-FRET tracer MAT379, that is >50-fold selective for *PfcProRS* over *HsProRS in vitro*. Thus, we describe a scaffold which offers a novel route to biochemical selectivity. These findings, together with our assay technology, will not only support our own efforts, but will also guide the development of other programs.

4) There are references explaining how halofuginone-resistant parasites gain resistance at molecular level. However, there are no efforts to utilize the information or rationally design the compounds based on these informations. At least, the authors should describe the experimental results based on the resistance mechanism and then further suggest the next direction to avoid the resistance mechanism.

We respectfully disagree with these statements. Our previous work identifying *PfcProRS* as the functional and molecular target of halofuginone and our discovery of altered proline homeostasis as the underlying mechanism of halofuginone tolerance/resistance have both motivated the research and informed the strategy presented in this manuscript. This is stated and referenced accordingly in our original manuscript (References 8, 16, and 27). We specifically note how this information is used in our approach (see lines 115-119 “Notably, some of these ligands, including T-3767758 (2) (Fig. 1a), displayed proline-uncompetitive steady state kinetics for *HsProRS*. We hypothesized that this property could be highly desirable for halofuginone-tolerant strains with elevated intracellular proline (halofuginone-induced), as it would potentially allow overcoming or selecting against this resistance mechanism.”).

We then proceed with our experimental results that validate this hypothesis. We first demonstrate the lack of cross-resistance to T-3767758, NCP26, and other analogs against the two halofuginone-resistant parasite lines that we previously identified (halofuginone-induced and HFGR-I), compared to the wildtype parent strain (Dd2-2D4). We also note that we undertook experiments that showed that NCP26 does not induce short-term resistance in *Plasmodium* parasites (Line 140: “Moreover, unlike

halofuginone, NCP26 did not induce rapid resistance in wild-type parasites that were subjected to intermittent drug pressure.”, and Supplementary Information Lines 288-294).

We specifically tested for the effect of proline on the binding affinities of the test compounds, which is why the TR-FRET-based ligand displacement assay was critical (as noted by this Reviewer in Major Comment #6). Unlike an enzymatic activity-based assay that requires proline because it is a ProRS substrate, we were able to profile our compounds in the absence and presence of proline and ATP to determine the compounds' binding modes with respect to each substrate.

In our revised manuscript, we have extended the resistance selection studies that establish clear guidance on the inhibitor characteristics that are desirable to avoid resistance.

5) The tracer must show the constant TR-FRET ratio irrespective of the existence of substrates. In addition, the assay conditions have to reflect the physiological conditions since the test compounds are either competitive or uncompetitive inhibitors that are affected by substrates concentrations. How much ATP and proline exist in Plasmodium and human cells? Are the assay conditions able to properly reflect the physiological conditions? What is the K_D value of MAT379 to HsProRS in the presence of either proline or ATP? What is the K_D value of MAT379 to PfcProRS in the presence of either 1 mM ATP or 500 μ M proline (or at the physiological range of the substrates concentrations). The authors suggested that the K_D -values for ATP and proline are 892 μ M and 457 μ M, respectively. Is the physiological range of the substrates concentrations similar to those K_D values?

In TR-FRET readouts, the detection settings (e.g. delay, integration time, gain) are optimized independently for donor and acceptor signal, and for the specific assay set-up (e.g. tracer concentration). Consequently, TR-FRET ratios are arbitrary values that are background corrected and may be normalized to negative and positive controls. It is therefore not expected for the TR-FRET ratios to be constant.

Even under identical settings, one would not expect the TR-FRET ratio to be constant in the presence and absence of substrate. For substrates that compete with the TR-FRET tracer, as is the case here for MAT379 and both proline and ATP, one expects that less tracer binds in the presence of substrate, which results in a decreased TR-FRET ratio.

The selection of substrate concentrations was informed by the respective K_D -values of the individual substrates for PfcProRS as determined by ligand displacement with tracer MAT379. The numbers are somewhat arbitrarily selected as “round” values within a 20-60% range of the respective K_D -values. The objective is to select a substrate concentration range that would not result in strong competition with the tracer, but sensitively report on uncompetitive inhibitors. We have tested various concentrations and confirmed the reported conditions to be well suited as evident by the reported validation experiment with halofuginone and NCP26 (see Fig. 2).

In this context, we would like to emphasize that it is not necessary, and potentially detrimental, to use substrates at physiological concentrations, including for biochemical enzyme activity assays. This is well validated for kinase assay platforms. The homeostatic level of intracellular ATP ranges generally from 0.5 to 5 mM, depending on cell type. However, most enzymatic kinase assays use ATP in the low micromolar range (e.g. PMID: 19662101). The specific ATP concentration is informed by the K_M -value

for the respective kinase of interest (“... the substrate concentration should be kept at $\leq K_M$ to ensure the sensitive detection of substrate competitive inhibitors, if desired.” (NIH Assay Guidance Manual NBK53196)). By definition ($K_M = V_{max} / 2$), higher concentrations have little impact (≤ 2 -fold) on the overall velocity but can significantly decrease the sensitivity for ATP-competitive ligands, which make up most kinase inhibitors. Predicted cellular activity can be directly inferred from the Cheng-Prusoff equation (we also note that any analysis needs to take the cellular kinase concentration into account as the limiting factor if the concentration of target protein exceeds the K_i -value; in this regard, please also see comments to Minor Comment 6).

For TR-FRET-based ligand displacement assays, or conceptionally analogous ligand displacement assay platforms based on fluorescence polarization or fluorescence lifetime (PMID: 17644772) readouts, different considerations apply. Unlike assays that measure steady state kinetics based on enzymatic turnover and therefore require the presence of substrate(s), ligand displacement assays are generally performed in the absence of substrate(s) as the tracers generally compete with substrate. Rather, ligand displacement assays measure ligand binding at thermodynamic equilibrium (i.e. the tracer takes the position of the substrate; in enzyme activity assays, the substrate is displaced by the inhibitor whereas in ligand displacement assays, like our TR-FRET assay, the tracer is displaced by the inhibitor OR by competitive substrates). For similar reasons, substrates are generally excluded from biophysical methods for ligand binding measurements, such as isothermal titration calorimetry (ITC) or surface plasmon resonance (SPR). As before, this is well established for kinase assays (PMID: 19564447), for which many ligand displacement assays are commercially available (e.g. see <https://www.thermofisher.com/us/en/home/industrial/pharma-biopharma/drug-discovery-development/target-and-lead-identification-and-validation/kinasebiology/kinase-activity-assays/lanthascreeentm-eu-kinase-binding-assay/lanthascreeentm-eu-kinase-binding-assay-validation-table.html>).

Substrates, as detailed above, are only included to detect uncompetitive ligand binding (e.g. PMID: 22453235. For ITC and the specific example of *PfcProRS*, see Jain et. al 2015 (PMID: 25817387). Also see PMID: 29497037 and http://tools.thermofisher.com/content/sfs/posters/MIPTTECH_2009_allosteric_inhibitors_KBA.pdf).

Furthermore, as noted above, the physiological concentration of ATP in human cells ranges from 0.5 to 5 mM, and 2.5 mM has been reported for *P. falciparum* (PMID: 10559194). The intracellular concentration of proline in human cells is 0.8 mM (PMID: 4829908). Unfortunately, we were unable to find reports providing quantitative information for the proline concentration in *P. falciparum*.

It is important to appreciate the limitations when comparing K_D values, a thermodynamic parameter that reports the true binding affinity of a ligand for the target protein, with a K_M value, which is a kinetic parameter that combines the substrate binding affinity and the turnover number (k_{cat}). Considering that $K_M / K_D = 1 + (k_{cat}/k_{off})$, then K_D is always $\leq K_M$ and only for $k_{off} \gg k_{cat}$ is K_D equal to K_M .

Heacock et al. have previously reported K_M values for ATP and proline for *HsProRS* (<https://doi.org/10.1006/bioo.1996.0025>). Comparing these K_M values to our experimentally determined K_D values provides further validation.

HsProRS: $K_{M, ATP} = 500 \mu M$ vs $K_{D, ATP} = 30.6 \mu M$ and $K_{M, Pro} = 180 \mu M$ vs $K_{D, Pro} = 67.1 \mu M$

6) The authors determined the K_D values in three different conditions, 0 μM Pro/ATP, 100 μM Pro and 500 μM ATP. Although this approach seems to be useful to study the MoA of the inhibitors, there are both proline and ATP in the actual working conditions of the inhibitors. The use of both proline and non-hydrolysable analog of ATP at the same time would derive much closer values to induce cytotoxic effect to the parasite/host and provide the direct evidence of parasite/host selectivity as the authors reasoned (Line 397, 398).

Yes, we agree with the Reviewer regarding the relevance of the presence of substrates in cellular context and the impact on selectivity. This is a critical point we noted in our original submission, and which we discuss in greater detail in our revised manuscript.

This example also illustrates the advantage of our TR-FRET-based ligand displacement assay platform that enables the quantitative determination of the various binding parameters (equilibrium dissociation constants (K_D), dissociation rates (k_{off}), and association rates (k_{on})), which would be difficult to accomplish with other assay setups.

Notably, our strategy enables not only the determination of inhibitor affinity in the absence and presence of substrates, but also the affinity of ATP and proline alone.

As the Reviewer correctly notes, the extent of ProRS inhibition will be different in the presence of competing substrates. However, the extent of inhibition not only depends on the absolute concentrations of the substrates, but also on the affinity of the substrates for the respective enzyme. Again, the kinase literature offers critical insights as exemplified for ATP-competitive JAK isoform-specific inhibition (PMID: 24814050, 23340136). Illustrated in the referenced example, many JAK inhibitors exhibit an apparent disconnect between biochemical inhibition and cellular activity. As shown, this was the result of equating biochemical with cellular potencies without considering differential substrate affinities. The biochemical data in the reported study were obtained at ATP concentrations equivalent to the respective K_M -value of the individual JAK isoforms, which varied >10-fold between the isoforms. However, since the cellular ATP concentration is the same for all enzymes, isoforms with low K_M -values (higher affinity) were less efficiently inhibited in cells by inhibitors that are supposedly equipotent in biochemical assays. The authors also demonstrate that correlating biochemical activities that are obtained using K_M concentrations of ATP and then corrected according to Cheng-Prusoff (as we discuss) provides a much more accurate prediction of cellular activity.

Similar considerations apply here. NCP26 is ATP-competitive and binds the host and parasite ProRS paralogs with comparable affinity to one another in the absence and presence of proline. However, HsProRS has much higher affinity for ATP, and given the similar cellular ATP concentrations in host and parasite, will be less sensitive to NCP26 in cellular context.

The predictive value of our biochemical assay is also illustrated by the comparative analysis of biochemically affinity and anti-parasitic activity across many inhibitors, which demonstrated high correlation (see SI Fig. 7). Thus, the K_D -values from our TR-FRET-based ligand displacement assay accurately predict cellular activity.

For these reasons and the considerations discussed under Major Comment #5, we do not believe that the proposed experiments using substrate combinations are advisable. Furthermore, assuming that

the non-hydrolyzable ATP analog is a suitable substitute that binds with comparable affinity to ATP, this approach could circumvent the formation of prolyl-AMP. However, for the same reason, it would not reflect all cellular contributions as it does not account for prolyl-AMP that is formed as an intermediate or for the ultimate product, prolyl-tRNA. The prolyl-AMP intermediate would be expected to exhibit a K_D -value comparable to ProSA, which we show to bind *PfcProRS* with low to mid-picomolar affinity.

7) How does MAT379 compete with proline? MAT379 is derived from T-3767758 and even the pyrazinamide-proline hybrids (MAT334 and MAT345) were found to be not competitive to proline. Is there any theoretical evidence for that? The graph for the determination of K_D values for ATP and proline of *HsProRS* should be provided in the Results section.

Before addressing the specific question, we want to ensure that we share the same terminology. It is important here to not conflate (which, unfortunately, is often done in the kinase inhibitor field) the concepts of competitive, uncompetitive, and noncompetitive inhibition with those of orthosteric and allosteric inhibition. An allosteric inhibitor is defined as a ligand that binds the target protein by occupying physical space that does not overlap with the space occupied by the respective substrate. Generally, this refers to a distal pocket, however, in a wider definition, may also include an adjacent pocket within the extended active site. Unlike orthosteric ligands, which can only be competitive since such ligands occupy the same physical space, an allosteric ligand can exhibit any inhibition mode (competitive, uncompetitive, or noncompetitive).

In support of these remarks, we would also like to draw the reviewer's attention to reference 24 (PMID: 27798837) in our manuscript. The authors identify glyburide and analogs as inhibitors of *PfcProRS* that bind an allosteric site adjacent to the active site. Importantly, despite binding to a distal site (as confirmed by crystallographic data), glyburide and related compounds demonstrate proline and ATP-competitive steady state kinetics (i.e. in a biochemical enzymatic activity assay). Similar considerations likely apply for the threonyl-tRNA synthetase inhibitor borrelidin, which has been identified as a "tri-competitive" inhibitor that competes with all three substrates, but does not occupy the ATP-binding site and instead occupies the threonine pocket, tRNA^{Thr} pocket, and an adjacent site. (PMID: 25824639).

In this regard, we would also like to further emphasize that neither T-3767758 nor NCP26 form any interactions (direct or water-mediated) with proline in their respective crystal structures bound to *HsProRS* (PDB: 5VAD, 7BBU) and *PfcProRS* (PDB: 6T7K). Unlike the tight interaction between halofuginone and ATP, which includes multiple hydrogen bonds and provides a convincing rationale for the ATP-uncompetitive mode of inhibition of ATP, it is more surprising to observe proline-uncompetitive binding for T-3767758 and NCP26.

At no point do we claim that MAT379, MAT334, and MAT345 are not competitive with proline. In fact, the opposite is true, we specifically state that these compounds are proline-competitive. The evidence is provided by both biochemical and cellular data. The ligand displacement assay with MAT379 shows that binding is competed by proline for *PfcProRS* and *HsProRS* (For *PfcProRS*, see line 202-203: "Dose-titration of ATP and proline revealed that MAT379 is not only ATP-competitive, as expected, but, unlike compound 2 or NCP26, also proline-competitive..." and for *HsProRS*, see lines 248-249: "As

expected, we found MAT379 to be both ATP ($K_D = 30.6 \mu M$) and proline ($K_D = 67.1 \mu M$) competitive.”). Thus, the requested data for the determination of the K_D -values for HsProRS with ATP or proline respectively are already in the results section and are also already included in Table 1 and the complete dose-response plots are shown in S.I. Figure 6 along with the corresponding PfcProRS plots. If binding was noncompetitive (as opposed to uncompetitive) with MAT379, we would not observe differential activity. In other words, dose titration of proline against a constant concentration of proline-noncompetitive tracer would yield a flat line. If binding was uncompetitive with proline, we would observe increased affinity, i.e. dose-titration of proline would not result in displacement of a proline-uncompetitive tracer (lower signal) but rather increased binding of the tracer (higher signal).

Furthermore, we demonstrate that the activity of MAT334 and MAT345 is reduced in halofuginone-resistant parasite strains (halofuginone-induced and HFGR-1), which have increased intracellular proline levels. The 20-to-40-fold decreased activity directly correlates with the ~20-fold-increase in proline in these parasite lines (Ref. 27).

We appreciate that at first glance, our co-crystal structures of the ternary complexes of MAT334 or MAT345 with PfcProRS and proline may mislead a reader to arrive at the conclusion that both ligands are not competitive with proline. After all, the structures show both ligands bound to the target enzyme without an obvious steric clash with proline or the distortion of the protein itself (compared to the respective structure with NCP26). To draw attention to this apparent contradiction in our data, we specifically included following statement in our original manuscript *“Notably, despite the proline-competitive mode of inhibition (Supplementary Fig. 8), we only obtained co-crystal structures for compounds MAT334 and MAT345 with PfcProRS in the presence, but not in the absence, of proline (PDB 7QC2 and PDB 7QB7).”* (Lines 292-295).

However, based on the reviewer’s comment, we realize that this may have been insufficient to provide adequate clarification and the wording could have been misunderstood. Because of the proline-competitive nature of MAT334 and MAT345, we initially only attempted to crystallize PfcProRS with both ligands in the absence of proline. However, crystallization efforts failed under all tested conditions. Unfortunately, it is not uncommon that certain protein-ligand conditions fail to provide crystals suitable for structural analysis and this does not simply reflect binding affinities, as evidenced by the similar failure to crystallize PfcProRS with NCP26 in the absence of proline even though NCP26 binds PfcProRS with higher affinity in the absence of proline than MAT334 does in the presence or absence of proline). We then attempted experiments in the presence of proline, which yielded the results reported in the manuscript. We note that under these conditions, both proline and MAT334/345 are present at millimolar concentrations. However, the presence of proline also requires to MAT334/345 to bind with (likely much) lower affinity, which can still be achieved given the millimolar concentration of the inhibitor. Together, our data suggest that in the absence of proline, MAT334 and MAT345 can engage in more productive, i.e. higher affinity, interactions with PfcProRS.

Conveniently, Van Aerschot et al. recently published co-crystal structure of human ProRS in complex with only proline (PDB: 7OSY), enabling the direct comparison to the apo structure (PDB: 4K86) and the co-crystal structures with proline and T-3767758 (PDB: 5VAD), providing valuable insights. Comparative analysis of these structures shows that binding of proline alone causes significant changes in ATP binding site and results in the reorientation of the side chains to more closely resemble the orientation found in the pyrazinamide co-crystal structures than that observed in the

apo-enzyme. This establishes a good mechanistic rationale for the proline uncompetitive binding mode of pyrazinamide-derived inhibitors. However, binding of proline to the apo-enzyme also causes an even more pronounced structural change along the exit vector from the active site, resulting in substantial reorientation and repositioning of the respective amino acid side chains that are predicted to interact with MAT379, MAT334, and MAT345, but not NCP26. This observation further establishes a plausible molecular basis for the proline-competitive binding mode of these ligands.

While we were not able to obtain structures in the absence of proline for NCP26, MAT334, and MAT345, we trust that the reviewer will agree that these theoretical considerations and experimental data are consistent and convincing. We have also included a similar discussion in our revised manuscript.

Minor comments:

1. MAT334 and MAT345 are not proline-competitive inhibitors. The complex structure and biochemical assay results support that MAT334 and MAT345 are not proline-competitive inhibitors. Although the authors described “unsuccessful attempts (Line 307)” in the end of the paragraph, they should revise the manuscript to prevent the reader’s misunderstanding by emphasizing that the compounds are designed to be proline-competitive and proved to be not proline-competitive through the series of experiments.

This comment directly pertains to Major Comment 7. As detailed in our response above, MAT334 and MAT345 are proline-competitive, and we provide comprehensive experimental data in support.

2. Line 330: Presentation of the chemical structure of compound 40 (iso-MT436) will help the readers to understand the paper.

The structure of iso-MAT436 (40) was shown in the Supplementary Information (see pages 42, 156, 157, 158, & 159). We have now included the chemical structure in Figure 5 of our revised manuscript.

3. Which compound is better for parasite/host selectivity? The direct comparison of the representative compounds (including control compounds like halofuginone and compound 2) of each class on parasite/host selectivity will provide useful information to the readers. The additional column in the Table 1 about parasite/host selectivity will be helpful.

We already provided this comprehensive comparison in Supplementary Figure 7 (also Supplementary Figure 7 in our revised manuscript) of our previous manuscript.

4. Line 336: This sentence is ambiguous. The exact comparison statement will help the readers to understand the paper. (An overlay with the corresponding ternary complex of PfcProRS with halofuginone and AMP-PNP (PDB 4Q15) revealed a nearly perfect overlap with halofuginone)

We agree with the reviewer and have change the text to “An overlay of MAT436 bound PfcProRS with the corresponding ternary complex of PfcProRS with halofuginone and AMP-PNP (PDB 4Q15) revealed a nearly perfect overlap of the halofuginone moieties...”

5. Line 357: human hepatocyte liver cells → human hepatocytes or human liver carcinoma cells

We thank the reviewer for pointing this out and have changed it to human hepatocytes.

6. Line 651, Line 677: What is the meaning of “these conditions” in the following sentences?

(Line 651: Compounds marked with ** were titrating ProRS under these conditions.

Line 677: Compounds marked with * are titrating HsProRS under these conditions.)

“These conditions” refers to the assay conditions, specifically the concentrations of PfcProRS/HsProRS as indicated for each specific panel in the same figure legend. Enzyme activity and ligand displacement assays fail to provide accurate K_D /IC₅₀ values when the K_D -value of the ligand is \ll [protein] (PMID: 20132208). This is because the ligand will titrate the enzyme. The IC₅₀ value measures the concentration at which 50% of the enzyme is in the ligand bound state (50% must be inhibited to reduce the signal by 50%). E.g. if 100 nM enzyme is used in a particular assay and two inhibitors with K_D -values of 0.01 and 1 nM, respectively, are studied in dose response, both compounds would yield quasi identical dose-response curves with an inflection point at ~50 nM (exact value will depend on substrate/tracer concentration) and appear equipotent. For this reason, we had to use lower enzyme concentrations in order to obtain meaningful K_D -values for the high affinity ligands (Figure 2). While we were able to push the sensitivity limit of the assay to accurately profile mid-picomolar ligands, we advocate low nanomolar enzyme concentration for standard profiling/screening experiments because of the excellent robustness. We therefore report both conditions with the titrating conditions marked with asterisks for clarity, which also serves to emphasize the often-overlooked issue of target titration.

7. Line 675: 2.5x KD \neq 0.15x KD

We thank the reviewer for alerting us to this mistake. The text should read “0.15x K_D ” and we have corrected this mistake in the revised manuscript.

8. The methods for determining the dissociation rates and association rates seem to be the same. What’s the difference?

Before answering the question, we would like to note two mistakes that we identified in the kinetics methods section (SI Methods lines 217-241) that have been corrected in the revised manuscript. We apologize for this mistake and any confusion they may have caused. 1) The experimental procedure for measuring dissociation kinetics stated that 10 μ M ProSA (or DMSO vehicle) was present in the 45 μ L diluent. However, the reagents were included in the initial 5 μ L sample with the concentrated ProRS and MAT379 to block MAT379 from binding to ProRS in the background control (note that 10 μ M ProSA = \sim 20,000 x K_D). 2) Equation 5 (Line 240) used to calculate the $k_{on,calc}$ should read “ $k_{on,calc} = k_{off} / K_D$ ” (not “ $k_{on,calc} = K_D \times k_{off}$ ”). This typo does not change the results since correct equation was used for data analysis.

The methods used are based on well-established experimental techniques (PMID: 20132208). Both involve rapid 10-fold dilution of ProRS followed by kinetic measurement of the TR-FRET ratio but differ in the conditions at $t = 0$ and whether ProRS and MAT379 were pre-incubated.

Dissociation rate (k_{off}): Concentrated ProRS (100 nM) is pre-incubated with \sim IC₈₀ MAT379 without competitor to ensure high tracer occupancy. The same conditions with 10 μ M ProSA serve as background control (no tracer bound). Rapid dilution into buffer will result in a tracer concentration $<$

IC₅₀ and consequently the system will equilibrate with dissociation rate constant (k_{off}) to a state with lower target occupancy. Using MAT379 at \sim IC₈₀ was chosen because this has been previously shown to provide a wide dynamic range which allows for more accurate kinetics measurements (PMID: 20132208).

Association rate constant (k_{on}): The rate of association can be derived directly, by measuring the time dependent formation of the tracer-target complex following addition of tracer to the target protein, or calculated ($k_{on} = k_{off} / K_D$) if the dissociation rate constant (k_{off}) and the equilibrium binding constant (K_D) are known.

We attempted to directly measure the association rate constant (k_{on}) by addition of ProRS (100 nM) that was pre-incubated with either DMSO vehicle or 10 μ M ProSA (to prevent association) to a solution containing MAT379 at concentrations $\gg K_D$ and measure the time dependent increase in TR-FRET signal. However, the association was too fast. We therefore derived k_{on} using the indirect method.

Reviewer #2 (Remarks to the Author):

Tye et al have prepared a well written manuscript that focusses on development of new inhibitors against malaria parasites. The experiment rationale and design are clear although the general ideas have been published earlier.

We appreciate that the reviewer found our manuscript to be well written and the experimental strategies to be clear. However, we are uncertain what the reviewer means by “the general ideas have been published earlier”, which suggests that our findings in essence have already been reported elsewhere.

We do agree that *Pfc*ProRS is recognized as an attractive target for malaria drug development, which was established by our research that identified ProRS as the functional and molecular target of halofuginone in humans and *Plasmodium* (References 8 and 14). This is the motivation of our study.

We also agree, that identifying compounds that are not cross-resistant with halofuginone resistance mechanisms is highly desirable (which is also based on our findings – see references 8, 16 (now 17), and 27). However, there are no previous reports on ProRS inhibitor classes that are demonstrably not cross-resistant to halofuginone.

Furthermore, we agree that triple-site aminoacyl-tRNA synthetase (aaRS) inhibitors have been postulated. However, despite extensive efforts, to the best of our knowledge no such inhibitors have been identified for any aaRS, including ProRS.

Finally, we agree that TR-FRET-based ligand displacement assays are widely used in biomedical research owing in large part to their superior performance to other biochemical assay technologies. Although the need for improved assay platforms for aaRS enzymes has been widely recognized, no successful efforts have been reported for any aaRS.

Although we agree that these general ideas have been postulated in the literature, they only define existing challenges and not resolved problems.

Some noteworthy results include:

1. Discovery of novel ATP mimetics that are proline uncompetitive based on pyrazinamide-based inhibitors that stem from previously published work from Takeda Pharmaceuticals where they discovered ATP-based inhibitors of PRS.
2. Discovery of a novel triple site inhibitor for PRSs that is essentially an ATP mimetic moiety linked to a halofuginone-quinazolinone moiety by a sulfamoylcarbamate group.
3. Parasite and host cellular inhibition tested against different cell-lines (Dd2-2D4, HFG-induced and HFGR-I Pf cell lines, Pb liver stage tissue culture and HuH7 host cell line) are described with these compounds.
4. A TR-FRET assay platform has been described but whether the reagents used are in public domain and/or accessible to all is unclear.

Consistent with the journal policies, the commercial sources and/or protocols for preparing all reagents used have been described in detail in the Supplementary Methods/Information or are appropriately referenced. We have previously reported the detailed synthetic protocols for the CoraFluor-1-Halo and CoraFluor-1-Pfp TR-FRET donor reagents in Ref. 32 (PMID 34675420). The detailed synthetic protocols for the novel small molecule ligands used in this study, including TR-FRET tracer MAT379, are included as supporting information. For the previously published small molecule ligands (reference compounds), the commercial source is noted or the appropriate literature reference(s) are cited. The expression plasmids are also described in detail in the Supplementary Methods (Lines 55-61) and will be deposited to Addgene following acceptance of the manuscript. The detailed protocols used for the transformation of these plasmids and subsequent expression, purification, and characterization of ProRS constructs for this assay platform are described in the Supplementary Methods (Lines 62-98). The detailed protocol for covalently labeling the HaloTag fused to ProRS with CoraFluor-1-Halo is described in the Supplementary Methods (Lines 116-134). As described in the Supplementary Methods (Lines 136-141), the unlabeled anti-His antibody is commercially available (Abcam Ab18184) and was labeled with CoraFluor-1-Pfp per the detailed protocol we previously reported in Ref. 32 (PMID 34675420). We therefore would like to emphasize that the reagents and strategies employed here will be accessible to both academic and industrial groups.

This work will further efforts for development of selective triple site inhibitors of PRS but as yet this has not been achieved, thus limiting the impact of this work.

The reviewer is correct that our newly-developed triple-site inhibitors do not exhibit biochemical selectivity for *PfcProRS* over the human paralog in the absence of ATP and proline. However, we would like to point out the considerations discussed in response to Major Comments #5 and #6 by Reviewer 1. Because of the differential substrate affinities for the human ProRS paralog it is predicted for MAT436 to be parasite selective. In fact, we demonstrate that MAT436 exhibits high *in vitro* selectivity for *Plasmodium* parasites over host cells, which is ultimately the most critical metric.

We pursued such inhibitors to experimentally test the hypothesis that triple-site inhibitors, which have previously been elusive, would offer superior potency as a means to expand the therapeutic window. While we were able to show that these compounds indeed exhibit high biochemical and cellular potency (mid-picomolar binding affinities and low-nanomolar *Plasmodium* growth inhibitory activities), we also reported that they suffer comparably reduced activity in halofuginone-resistant parasites (halofuginone-induced and HFGR-I), which is ultimately undesirable, yet the extent was not expected.

Some major issues that need to be addressed:

1. For the KD experiment, why were these concentrations (100 μ M L-pro or 500 μ M ATP) chosen for TR-FRET? Is it just to have excessive natural substrate or any other reasoning exists? And if it is for saturation – why were these exact particular values chosen? Is there any correlation with the substrate K_m here? Considering these scaffolds are competitive with either of the natural substrates, the relative concentrations of the natural substrate and the enzyme might have differential effects on the measured KD.

The general concerns here mirrors those discussed in Reviewer 1's Major Comments #5 and #6. We would therefore like to refer the reviewer to the discussion above. We are confident that it fully addresses all concerns and questions.

2. Furthermore, KD should have been determined in another combination as well – in presence of both natural substrates (L-pro and ATP at $\sim 2.5K_m$ concentration ranges as indicated in ref 22 “How to measure and evaluate binding affinities”) and in presence of tRNA^{Pro}. Currently, the data points only suggest a very good association between the protein and drugs, however, this does not categorically prove biochemical inhibition.

The reviewer raises several concerns here and while we believe that they are largely addressed by our response to Reviewer 1's Major Comments #5 and #6, we want to ensure that we are fully responsive.

We have carefully studied the reference and believe that the reviewer refers to Appendix 3 (“Protein concentrations ~ 2 – 5 times above the K_D for the labeled ligand ... should typically be used.”).

We would like to note that this guidance does not stringently apply to TR-FRET-based ligand displacement assays, as used here. The specific example in Appendix 3, as noted in the main text, provides an experimental solution to determine direct binding if other approaches have failed (see paragraphs immediately before the discussion in Ref. 22). Since our assay robustly detects binding for both isoforms, this issue does not apply. However, we are using MAT379 with *PfcProRS* already at $2.5 \times K_D$, while the same tracer is used with *HsProRS* at $0.15 \times K_D$. In this respect, we also would like to point the reviewer to PMID: 19564447 ("*The ability to use a sub- K_D concentration of kinase, as well as a concentration of tracer that is close to the K_D for the kinase-tracer interaction, is particularly advantageous when using tracers that have relatively low affinity for the kinase ($K_D > 100$ nM) and is a distinguishing feature of the TR-FRET format that we have presented when compared to binding assays performed in FLT, FP, or EFC formats.*") for a practical application to kinase profiling platforms.

Care must be taken when comparing K_M , which is a kinetic constant, with a K_D value, which is an affinity constant (see Reviewer 1 Major Comment #5). However, as discussed above, it is not advisable to include the substrates at such high concentrations, since the tracer is competitive with ATP and proline. Furthermore, combining both substrates, would result in the formation of prolyl-AMP, which is structurally similar to ProSA, and would be expected to potentially inhibit MAT379 binding to ProRS with comparable affinity (low to mid-picomolar).

We are also uncertain why the reviewer considers including tRNA^{Pro} necessary. In fact, many widely used enzyme activity assays for ProRS (including those cited in ref 8 – Herman *et al.*, *Sci Transl Med* 2015; ref 15 – Zhou *et al.*, *Nature* 2013; ref 20 Shibata *et al.* *PLoS One* 2017; ref 21 Adachi *et al. Biochem Biophys Res Commun* 2017; and ref 26 Arita *et al. Biochem Biophys Res Commun* 2017) do not include tRNA in the assay setup. While some other reports utilize assay platforms that include tRNA, they mostly use tRNA fractions isolated from yeast or bacteria. Those that use specific tRNA^{Pro} isoacceptors, rely on *in vitro* transcribed tRNA^{Pro} that lacks the base modifications found in mature tRNA, which can modulate the interaction with the cognate aaRS, and should therefore only be considered as substrate substitute.

While we are aware that Jain *et al. Structure* 2015 report the use of *in vitro* transcribed tRNA^{Pro} in a biochemical aminoacylation assay, no information is provided on how the choice of substrate concentrations was informed (ATP = 200 μ M, proline = 5 mM, tRNA^{Pro} = 8 μ M), as no K_M values have been determined by steady state kinetics and no discussion is provided on how those can be considered as physiologically relevant. Furthermore, this assay does not detect the amount of aminoacylated tRNA^{Pro} but quantifies the amount of inorganic phosphate byproduct (as it is done in the tRNA-free biochemical assays). Therefore, tRNA in this experimental setup only functions as a "sink" to allow turnover. Notably, this assay specifies the use of 400 nM recombinant ProRS, but reports IC₅₀ values in the single and double digit nanomolar range, which, as discussed above, is beyond the theoretical sensitivity limit of such a setup and raises questions about how the authors obtained these specific values. Although we would prefer to not dive into this comparative analysis that more critically discusses these limitations in a revised manuscript, we are happy to do so if deemed relevant and necessary by the reviewer.

3. Enzyme inhibition assays are missing and hence there is no biochemical evidence (despite the structural evidence) of the mode of action of the compounds used in this study. The compounds do bind

(as per structures) to PRS but whether they actually inhibit the enzyme activity remains to be shown. The authors themselves realise that even a previous study (ref 20, 21, 26, 39) did not have these relevant data (Page 17 line 402-404 : “Although the authors did not provide direct biochemical evidence for PfcProRS inhibition, they demonstrated oral efficacy of their lead compounds in a murine model of human malaria, which showed equal potency as NCP26 in ABS parasites”. Biochemical inhibition data will be vital in going forward with these compounds and their derivatives as well as doing 3D structures of all possible hit compounds will be tedious.

According to the definition provided by NatureSpringer for their portfolio journals “A biochemical assay is an analytical in vitro procedure used to detect, quantify and/or study the binding or activity of a biological molecule, such as an enzyme” (<https://www.nature.com/subjects/biochemical-assays>).

Consistent with this definition, we provide unambiguous, quantitative biochemical evidence for ligand binding, including the mode of inhibition, at a resolution that has not been possible before.

This definition, however, does not change the validity of the referenced sentence in the comment (Page 17 line 402-404). This previous paper, which followed our work and cites our invention disclosure identifying the ATP-targeted pyrazinamides as the first such examples as being active against *P. falciparum*, did indeed not provide any form of biochemical or structural evidence.

Unfortunately, despite comprehensive biochemical evidence, which is supported by structural data for all ligand classes, that the presented ligands bind the active site, and the very high correlation between biochemical affinity and cellular potency (including the cross-profiling against halofuginone-resistant lines), the reviewer considers demonstrating enzyme inhibitory activity in a biochemical activity assay as definitive requirement to validate the proposed mode of action.

While we would be happy to perform such experiments if they could resolve this question, we note that none of the current enzyme activity assay platforms are expected to be capable of differentiating most of our compounds for the reasons noted above and in our manuscript. This shortcoming motivated the development of the TR-FRET-based ligand displacement assay, which, as discussed, is a widely accepted strategy for other protein families. In fact, Takeda was only able to get approximate binding affinities of their *HsProRS* inhibitors, including T-3767758 (compound 2), by employing an equilibrium binding experiment that estimates the fraction of bound enzyme by mass-spec quantification of denatured ligand-protein complexes.

In addition, following the same logic that the reviewer uses to reject our claim, demonstration of biochemical inhibition of enzyme activity could not serve as evidence for the *in vitro* molecular mode of action (i.e. that the compounds inhibit the growth of or kill *P. falciparum* parasites by inhibition of *PfcProRS* as principal target and do not exert their antiparasitic activity through another target). However, we are confident that the NCP26 resistance selection studies, which have been incorporated in the revised manuscript, unambiguously resolve this concern.

In addition:

1. Lines 75-77: “Crystallographic data of the co-complexes with human and Plasmodium ProRS revealed that halofuginone binds the A76-tRNAP^{ro} 76 and proline-binding pockets of the active site (Fig. 1a), which are highly conserved between both homologs.” The appropriate ref for this is Jain et al who resolved the PfPRS-HFG complex (PDB ID: 4YDQ) (Structure. 2015 May 5;23(5):819-829. doi: 10.1016/j.str.2015.02.011. Epub 2015 Mar 26).

We are grateful to the reviewer for pointing out this mistake and sincerely apologize for this oversight. The correct reference was included in our previous draft but accidentally deleted when finalizing the document. We have corrected this in our revised submission.

2. In supplementary Figure 1- the ribbon may be hidden for clarity. The surface transparency can be decreased to show the bound ligands clearly, particularly bound proline.

As suggested by the Reviewer, we have hidden the ribbon and increased the surface transparency to better show the ligands.

3. Line 695: Supplementary Table 1 was mentioned but this is missing in supplementary data.

We thank the Reviewer for pointing this out. Supplementary Table 1 was submitted as Excel file and should have been provided by the editorial office.

4. Supplementary Figure 2. It would be better for the reader to connect the dots if a similar annotation (circle/square/triangle) is used for the same compounds in different panels. That way a consistency and continuity would remain.

We have revised this figure to increase readability.

5. In Supplementary Table 2:

a) Space group notation – screw axis notation should be in subscript.

We have corrected the space group notation accordingly.

b) “ $I / s (I)$ ” should be replaced by “ $I / \delta I$ ”.

We suspect a font change may be altering exactly what the Reviewer wrote here. It is currently “ $I / \delta I$ ” (I over δI) in the version downloaded from Nature Communications website. However, the reviewer is correct, and we changed this to “ $I / \sigma I$ ” (I over σI) in our revised manuscript.

c) The symbol “” should be removed in “51.68-`1.79”.

We have corrected the typo in our revised manuscript.

d) Decimal approximation may be needed for the average B-factor values.

We now report 3 significant digits (i.e. 1 digit after the decimal).

6. Supp_Methods: Line 167: D should be subscript in KD, i.e. KD should be K_D .

We have corrected the typo in our revised manuscript.

Reviewer #3 (Remarks to the Author):

This is an interesting paper that greatly advances inhibitor work for *Plasmodium falciparum* (malaria) Prolyl-tRNA-synthetase (ProRS). The paper merges: 1) assay development of a unique FRET assay that can be used across many AARS targets that solves many issues with previous assays; 2) moving from a good industry lead to leads that are more potent; 3) using structural biology and enzyme assays to characterize the binding properties, including which of the 3 pockets (tRNA vs. Proline vs. ATP pockets); and finally, 4) using structure-guided drug optimization to make a unique binder of PfProRS that reaches into all 3 pockets. The paper is greatly aided by analysis of wild-type Pf parasites and parasites that have elevated proline levels and also a mutant PfProRS and parasites that express the mutant. It is clear that triple-pocket binding compounds are rendered ineffective by these mutant Pf strains. My only suggestion about improving the paper is that it took me 2-3 re-readings of

the paper to get the basic points. There are a lot of details in the main body of the paper that detract from readability. The authors could move some of the details into supplement, so it doesn't break up the flow of the paper. In my opinion, this would improve readability.

We would like to thank the Reviewer for the positive feedback and their time thoroughly reviewing the manuscript. We appreciate that our manuscript is detail-rich and have attempted to improve readability in our revised manuscript. In addition, we have split the original Fig. 3 into two parts (Fig. 4 and Fig. 5 in our revised manuscript).

REVIEWER COMMENTS

Reviewer #2 (Remarks to the Author):

There remain many gaps that reduce enthusiasm for this revised work. Several assays are still not covered as described below. Considering the stature of Nature Communications and the fact that such a journal publishes work that is effectively a leap in scientific understanding of a concept. This work shows the results of a variable assay platform with that of a highly optimised and sensitive assay that is more or less of the same fidelity with both PRS paralogs. A drug discovery or a protocols specific journal or one that is focussed on enzymology will be more suitable for this work.

1. The reliability and fidelity of the TR-FRET assay raises questions on the physiological relevance of the concentration regimes chosen for these assay – i.e., the ratio of target enzyme and the drug/substrate concentrations. Further the testing of affinity was done only in presence of individual substrates which too is physiologically irrelevant – in presence of both substrates would be necessary.
2. MAT379 as a tracer molecule is constitutively bound to the catalytic pockets and is replaced only when another drug with a higher affinity displaces it. Therefore, this study shows that these drugs inhibit cellular parasitic growth and that they bind to PRS in apo condition, in the presence of excess L-pro and in presence of excess ATP. However, no mention of binding in presence of both substrates at ~2.5-3x Kd is available – as is the physiological relevance.
3. To unambiguously prove enzymatic molecular inhibition – recombinant protein expression, activity standardization and consequent inhibition of the same in presence of the drugs and natural substrates would be required.
4. To conclude inhibition of PfPRS by these test compounds – enzyme activity inhibition profiles would be best.

Minor Comments

1. Abstract Ln 38 – this is not the first triple-site ligand – it might be the first synthetic one – Borrelidin is a known multiple site binder to aaRSs.
2. Ln 40 – “our data inform” should be “our data informs”.
3. Ln 42 – “promise” should be “promises”.
4. Ln 48 – it should be disease, not diseases.
5. Ln 183 – “inhibitors” should actually be “binders” – albeit at a better affinity than the natural substrates – but this is still affinity only.
6. Fig 2 panels should have the determined Kds of the species as an inlet within the graphs. Would read better for the reader.
7. Ln 274 – correlation does not suggest on target activity – it merely correlates. Correlation does not necessarily indicate causation. Nevertheless, a high correlation factor does indicate some degree of causation.
8. Fig 3b labels are blurry – please sort these out.
9. Ln 321 – maybe “prolyl-like” would better express what the authors wish to communicate.
10. Ln 374 – inhibitor should be binder – the crystal structure only unambiguously proves binding site and pose, not inhibition.
11. Ln 458 – triple site inhibition has been elucidated previously, a single chemical synthetic compound with triple site inhibition is an effort that is commendable. And gauging selectivity of compounds via mode of inhibition studies too.

Reviewer #3 (Remarks to the Author):

I think the revision is very comprehensive, the authors have addressed all the points of the 3 reviewers, and the paper is acceptable now.

Reviewer #4 (Remarks to the Author):

As requested by the editors, no formal and full evaluation of the paper is expected, but an evaluation of the rebuttal and an opinion on whether all remarks have been properly addressed. See notes to the editor.

Report:

Tye et al have prepared a well written manuscript discussing the development of new inhibitors against malaria parasites. This was accomplished by development of a new FRET assay, a strategy useful as well for development of inhibitors for other targets. They further used structural biology and structure-guided drug optimization to obtain a triple-site-binding inhibitor targeting all three substrate pockets (amino acid, ATP, and the tRNA terminal adenosine).

I do not fully agree with all claims of the authors, but overall the paper is highly instructive and especially after the present revision worthwhile publishing following some minor corrections. The rebuttal is very detailed and extensive as well and, in my opinion, correctly addresses most of the referee remarks. I will however comment a bit on some of these replies.

Reviewer #1 comments:

First general remark: "Although the authors showed the activities of the triple-site inhibitors for human and Plasmodium ProRSs, they did not show efficacy on halofuginone-resistant parasites". The rebuttal extensively answers this remark. It is correct that NCP26 and analogous inhibitors do not induce rapid resistance in wild-type parasites, and it has been clearly documented that such compounds have greatly reduced propensity to evolve resistance but it does not answer the specific question of reviewer #1 regarding the activities of the triple-site inhibitors and their efficacy on halofuginone-resistant parasites. The rebuttal focuses on NCP26 which is not a triple-site inhibitor, but indeed these analogues are acting through PfcProRS as the principle functional target in vitro, and are ATP-competitive, proline-uncompetitive PfcProRS inhibitors, without much resistance generation.

In contrast, for MAT334 and 345 although not being triple-site inhibitors the authors already argued (in answering remark 7) that they demonstrated that the activity of MAT334 and MAT345 is reduced in halofuginone-resistant parasite strains (halofuginone-induced and HFGR-I), which have increased intracellular proline levels. The 20-to-40-fold decreased activity directly correlates with the ~20-fold-increase in proline in these parasite lines (Ref. 27).

It would have been nice therefore to see the effect of increased proline concentrations on the activity of real triple site inhibitors. Now the paper only states (further down around line 464) that the triple-site compounds bound both ProRS paralogs with subnanomolar affinity in the absence of either substrate (proline or ATP). Their high biochemical affinity translated well into antiparasitic activity against both ABS and liver stage Plasmodium parasites. The latter conditions obviously do contain competing proline.

This first referee remark is followed by "7 major comments of reviewer #1" which in my opinion each time are adequately answered by an extensive rebuttal.

Only for the second paragraph of the rebuttal on remark 7 which discusses borrelidin not being a triple-site inhibitor, I have mixed feelings. The same statement is found on a number of places in the paper (in the abstract and within results as well as discussion) where the same is mentioned. The authors claim that borrelidin is a "tri-competitive" inhibitor that competes with all three substrates, but does not occupy the ATP-binding site. I however immediately recalled a Nature Communications paper of 2015 (DOI: 10.1038/ncomms7402) on borrelidin which claims the compound to bind all three substrate binding sites ["a single molecule of borrelidin simultaneously occupies four distinct subsites

within the catalytic domain of bacterial and human ThrRSs. These include the three substrate-binding sites for amino acid, ATP and tRNA associated with aminoacylation, and a fourth 'orthogonal' subsite created as a consequence of binding"]. The authors in their rebuttal refer to the same paper (PMID: 25824639). The same paper is referred to in the manuscript as well but only at the end (ref 49). We have to admit that the tRNA third site is only partially occupied by borrelidin in comparison with the new inhibitors of this manuscript, in which halofuginone is included in the new triple-site binding structures and therefore truly fully occupying all three substrate sites. I nevertheless am inclined to suggest the authors should slightly rephrase the various parts in the paper where the uniqueness of their triple-site inhibitors is mentioned. In my opinion, they should refer to the partial occupation of the third site by borrelidin.

In addition, all minor comments (1-8) have been adequately dealt with and I specifically support inclusion of the very instructive figure 5 for the revised version (minor comment 2).

Reviewer #2 comments:

"The experiment rationale and design are clear although the general ideas have been published earlier."

The authors disagree with this simple but a bit negative statement of the referee, and I support the authors in that this manuscript really offers several new findings, and attempts to formulate solutions for the existing challenges. The authors further adequately answer both objections formulated by this referee in "result 4". (reagent availability and limited impact of the work).

Indeed, development of selective triple site inhibitors of PRS has not been achieved yet, but high in vitro selectivity for Plasmodium parasites has been shown for MAT436. As pointed out above it would have been desirable to see analogous tests also with the real triple site inhibitors.

Further, all 3 major issues raised by referee #2 have been answered at length, and likewise all smaller remarks have been adequately dealt with.

Further personal comments:

* Line 120 in the revised manuscript states: "Although no synthetic protocols for compound 2 or analogs have been reported"

However, synthesis of the Adachi compound (compound 2) was reported by L. Pang et al. in 2021 (<https://doi.org/10.3390/ijms22157793>), a reference paper which is quoted by the authors themselves for their structural arguments in their rebuttal.

* Figure 5: MAT436 (33) should read compound #34 instead (typo)

REVIEWER COMMENTS

Reviewer #2 (Remarks to the Author):

There remain many gaps that reduce enthusiasm for this revised work. Several assays are still not covered as described below. Considering the stature of Nature Communications and the fact that such a journal publishes work that is effectively a leap in scientific understanding of a concept. This work shows the results of a variable assay platform with that of a highly optimised and sensitive assay that is more or less of the same fidelity with both PRS paralogs. A drug discovery or a protocols specific journal or one that is focussed on enzymology will be more suitable for this work.

We appreciate the continuous effort by this reviewer. We are sorry to hear that our previous response did not satisfy this reviewer. We sincerely hope that our response to this reviewer as well as the other reviewers below will satisfactorily address the remaining concerns.

1. The reliability and fidelity of the TR-FRET assay raises questions on the physiological relevance of the concentration regimes chosen for these assay – i.e., the ratio of target enzyme and the drug/substrate concentrations. Further the testing of affinity was done only in presence of individual substrates which too is physiologically irrelevant – in presence of both substrates would be necessary.

We believe that the concerns raised here and below by reviewer #2 have been addressed in great detail in our previous response (as agreed to by reviewers #3 and #4). Our approach and analysis are consistent with the NIH Assay Guidance Manual (<https://www.ncbi.nlm.nih.gov/books/NBK53196/>). However, the reviewer appears to not agree with the validity of this approach and the significant body of literature that has been published in this regard. We have supported all of our claims with specific examples published in topic-relevant peer reviewed journals. The reviewer makes several strong statements in opposition to our response and reiterates some of their previous concerns. We have unsuccessfully tried to identify literature examples that would provide a basis for the concerns raised by this reviewer. We would be grateful if the reviewer could provide relevant literature references that would allow us to better address their concerns.

The reviewer is concerned about the reliability and fidelity of our assay platform. Our assay has “ $Z' > 0.95$ at 500 pM HT-*PfcProRS*, $Z' = 0.71$ at 250 pM HT-*PfcProRS*, and $Z' = 0.60$ at 20 pM HT-*PfcProRS* supplemented with 1 nM CoraFluor-1-labeled anti-His6 antibody” (line 228-230) and “ $Z' = 0.80$ at 1 nM CoraFluor-1-labeled HT-*HsProRS* and $Z' = 0.76$ at 50 pM CoraFluor-1-labeled HT-*HsProRS* supplemented with 1 nM CoraFluor-1-labeled anti-His6 antibody” (line 244-246). According to Zhang et al. (ref. 34), an assay with $Z' > 0.5$ is considered excellent (with 1.0 representing a perfect assay). Based upon these statistical considerations, we are confident that our assay is sufficiently reliable.

The reviewer also raises concerns about “the physiological relevance of the concentration regimes chosen for these assay – i.e., the ratio of target enzyme and the

drug/substrate concentrations”. If the reviewer was indeed correct, these concerns would not only apply to our assay approach but similarly imply that any data generated with most TR-FRET assays, surface plasmon resonance assays, isothermal titration calorimetry assays, fluorescence polarization assays, or biolayer interferometry assays are physiologically irrelevant. By extension, structural biology data from techniques such as x-ray crystallography, cryogenic electron microscopy (cryo-EM), and nuclear magnetic resonance spectroscopy (NMR) would only be relevant if the studies are performed at physiological protein concentrations and at the physiologically relevant ratios of their natural substrates and ligands.

2. MAT379 as a tracer molecule is constitutively bound to the catalytic pockets and is replaced only when another drug with a higher affinity displaces it. Therefore, this study shows that these drugs inhibit cellular parasitic growth and that they bind to PRS in apo condition, in the presence of excess L-pro and in presence of excess ATP. However, no mention of binding in presence of both substrates at $\sim 2.5\text{-}3\times K_D$ is available – as is the physiological relevance.

We are unsure about the main argument raised in this comment and the intended meaning of “constitutively bound”, as it appears that the reviewer implies that the tracer would remain continuously bound to the target protein and only displaced by a competing ligand if such ligand exhibits greater affinity for the target.

However, neither statement can be correct. We have shown that MAT379 binds reversibly to *PfcProRS* or *HsProRS* and, in accordance with the literature (ref 33), determined its dissociation kinetics in order to determine the necessary minimum incubation period for our TR-FRET assay: “To determine the minimum incubation time required for equilibrium conditions (five half-lives), we measured the first-order dissociation rate constant (k_{off}) for MAT379 by 10-fold dilution of an equilibrated solution of CoraFluor-1-labeled HT-*PfcProRS* (100 nM) and $\sim EC_{80}$ MAT379 (560 nM) which yielded k_{off} -value of $<0.16 \text{ min}^{-1}$ (Supplementary Fig. 4c-d), suggesting that quasi-equilibrium is reached within 15 min (unless the test compounds themselves exhibit slow binding kinetics).³³” (line 193-198) We similarly determined that for MAT379 and *HsProRS*, $k_{\text{off}} = 0.529 \text{ min}^{-1}$ (Supplementary Fig. 4c-d).

As such, there is a dynamic equilibrium of MAT379 binding and dissociating from ProRS. A competing ligand would not replace the tracer but bind to a free protein, preventing the binding of MAT379.

It is also not correct that only higher affinity ligands (i.e. $K_D, \text{ligand} < K_D, \text{tracer}$) can compete with the tracer. Basic thermodynamic considerations dictate that ligands with lower affinity can compete as well. However, to accomplish comparable competition requires a higher concentration of the low-affinity ligand. In fact, we have exploited this principle for the determination of the K_D -values of ATP and proline, which, as an example, are respectively $\sim 9,000$ -fold and $\sim 4,500$ -fold lower for *PfcProRS* than MAT379 (Figure 2c, Supplementary Figure 5, and Supplementary Table 2).

As we have already discussed in our previous response, it is not possible to perform this assay in the presence of both substrates at $2.5\text{-}3\times K_D$ as they would

react with each other to form prolyl-AMP as an intermediate. For this very reason, these conditions are not utilized in the literature when isothermal titration calorimetry (ITC) is used for the determination of binding affinity.

We therefore believe that the concerns raised here have been satisfactorily addressed in our previous response (as agreed to by reviewers #3 and #4). However, should the reviewer not be persuaded by our reasoning and the literature references that have been provided in support of our previous response, we likewise ask the reviewer to support their opposition with appropriate literature references.

3. To unambiguously prove enzymatic molecular inhibition – recombinant protein expression, activity standardization and consequent inhibition of the same in presence of the drugs and natural substrates would be required.

4. To conclude inhibition of PfPRS by these test compounds – enzyme activity inhibition profiles would be best.

Re 3 and 4: In response to the concerns raised following the previous submission, we included extensive resistance selection studies in the revised manuscript that unambiguously have proven that *PfcProRS* is the molecular target of the pyrazinamide-based inhibitors. In addition, we want to reiterate that inhibition of the enzymatic activity of human ProRS by Compound 2 and other pyrazinamide analogs has been previously demonstrated in the referenced literature.

In agreement with the editor, the requested enzymatic activity assays are therefore not required.

(Reviewer #2) Minor Comments

1. Abstract Ln 38 – this is not the first triple-site ligand – it might be the first synthetic one – Borrelidin is a known multiple site binder to aaRSs.

This comment is similar to Minor Comment #11.

We address both comments comprehensively below in response to reviewer #4 Comment #4.

2. Ln 40 – “our data inform” should be “our data informs”.

We understand that there is a continuous and often heated debate over the use of “data” as plural or singular noun that extends beyond the scientific literature (e.g. see <https://www.economist.com/graphic-detail/2012/07/13/data-or-datum> and <https://www.wsj.com/articles/BL-REB-16665>). Linguistically, “data” is the plural form of “datum”. In common language (in particular, American English), “data” is frequently used as singular noun and many publishers have agreed to accept both forms. However, British publications, including SpringerNature publications, tend to exclusively use “data” as a plural noun. Accordingly, we have followed the grammatical style of *Nature Communications* in our manuscript. For the

convenience of the reviewer, we provide some recent examples in *Nature Communications*.

- <https://www.nature.com/articles/s41467-022-30469-3> (e.g. Figure 2 legend)
- <https://www.nature.com/articles/s41467-022-30264-0> (e.g. Figure 2 legend)
- <https://www.nature.com/articles/s41467-022-30429-x> (e.g. Figure 2 legend)
- <https://www.nature.com/articles/s41467-022-30530-1> (“New data are then assigned...”)

4. Ln 48 – it should be disease, not diseases.

We thank the reviewer for pointing out this typo.

5. Ln 183 – “inhibitors” should actually be “binders” – albeit at a better affinity than the natural substrates – but this is still affinity only.

In our previous response, we provided an extensive discussion on why an orthosteric ligand is always a competitive inhibitor. Otherwise for an enzyme to catalyze a given reaction the ligand and substrate would have to occupy the same physical space at the same time. Our structural data for all inhibitor classes unambiguously show that they bind orthosterically with respect to ATP (and for MAT436, also proline and A76-tRNA^{Pro}). We are unaware of any example for a small molecule ligand that binds a substrate pocket in the active site of an enzyme without inhibiting the catalytic activity of this protein. We would be grateful to the reviewer to please provide us with one or more representative literature examples for any such instance to include as references in our manuscript.

6. Fig 2 panels should have the determined K_D s of the species as an inset within the graphs. Would read better for the reader.

While we appreciate the suggestion, we would like to note that this is not standard practice in *Nature Communications* for K_D values or other constants determined from dose-response data (e.g. IC_{50} and EC_{50}) as seen in the following examples from the most recent issue of *Nature Communications*:

- <https://www.nature.com/articles/s41467-021-22235-8#Fig2>
- <https://www.nature.com/articles/s41467-022-30430-4#Fig5>
- <https://www.nature.com/articles/s41467-022-30338-z#Fig2>

However, we would be happy to revise the figure accordingly should the editorial team find that the figure would benefit from the addition of such an insert.

7. Ln 274 – correlation does not suggest on target activity – it merely correlates. Correlation does not necessarily indicate causation. Nevertheless, a high correlation factor does indicate some degree of causation.

We fully agree with the reviewer that correlation does not imply causation. However, at no point have we made such a claim. In the specific section of our manuscript, we wrote: “The strong correlation ($r_s = 0.85$) between the biochemical activity of this inhibitor set and the cellular potency against asexual blood stage *P. falciparum* strongly suggests that the antiparasitic activity of pyrazinamide analogs is the direct consequence of on-target activity.” We did not state that the correlation “implies causation”. We used “suggests”, which is standard phrasing

to express the strong association between two variables and enables the formulation of our hypothesis.

In fact, the sentence that follows directly after “To experimentally validate *PfcProRS* as the principle functional target and ...” addresses this very point, namely that the correlation is no evidence, and to provide experimental evidence we conducted the resistance selection experiments.

8. Fig 3b labels are blurry – please sort these out.

This issue was caused by the conversion of the provided master files and has been corrected.

9. Ln 321 – maybe “prolyl-like” would better express what the authors wish to communicate.

We appreciate the suggestion. However, “prolyl” is the accurate nomenclature for the (S)-pyrrolidine-2-carbonyl moiety that we are describing. We refer the reviewer to the following chemical databases:

- <https://pubchem.ncbi.nlm.nih.gov/compound/57461922>
- <https://www.ebi.ac.uk/chebi/searchId.do?chebiId=26274>

In fact, using “prolyl-like” would suggest that the moiety is similar to but not identical to the presented structure, which would be inaccurate and misleading.

10. Ln 374 – inhibitor should be binder – the crystal structure only unambiguously proves binding site and pose, not inhibition.

We believe the term “inhibitor” to be justified and accurate based on the anti-proliferative activity of this compound against *Plasmodium*. The use of the term “inhibitor” in this context is also consistent with common practice of RCSB PDB.

11. Ln 458 – triple site inhibition has been elucidated previously, a single chemical synthetic compound with triple site inhibition is an effort that is commendable. And gauging selectivity of compounds via mode of inhibition studies too.

This comment is similar to Minor Comment #1.

We address both comments comprehensively below in response to reviewer #4 Comment #4.

Reviewer #3 (Remarks to the Author):

I think the revision is very comprehensive, the authors have addressed all the points of the 3 reviewers, and the paper is acceptable now.

We would like to thank the reviewer for their time and feedback. We are glad that we were able to address the previous concerns and we appreciate that the reviewer also evaluated the critiques of the other reviewers and our responses to these comments.

Reviewer #4 (Remarks to the Author):

As requested by the editors, no formal and full evaluation of the paper is expected, but an evaluation of the rebuttal and an opinion on whether all remarks have been properly addressed. See notes to the editor.

Report:

Tye et al have prepared a well written manuscript discussing the development of new inhibitors against malaria parasites. This was accomplished by development of a new FRET assay, a strategy useful as well for development of inhibitors for other targets. They further used structural biology and structure-guided drug optimization to obtain a triple-site-binding inhibitor targeting all three substrate pockets (amino acid, ATP, and the tRNA terminal adenosine).

I do not fully agree with all claims of the authors, but overall the paper is highly instructive and especially after the present revision worthwhile publishing following some minor corrections. The rebuttal is very detailed and extensive as well and, in my opinion, correctly addresses most of the referee remarks. I will however comment a bit on some of these replies.

We thank reviewer 4 for their time and effort to provide a rapid review. We hope that our point-by-point responses to their comments will alleviate any remaining concerns with our claims.

Reviewer #1 comments:

First general remark: "Although the authors showed the activities of the triple-site inhibitors for human and Plasmodium ProRSs, they did not show efficacy on halofuginone-resistant parasites".

(Reviewer #4 – Comment #1) The rebuttal extensively answers this remark. It is correct that NCP26 and analogous inhibitors do not induce rapid resistance in wild-type parasites, and it has been clearly documented that such compounds have greatly reduced propensity to evolve resistance but it does not answer the specific question of reviewer #1 regarding the activities of the triple-site inhibitors and their efficacy on halofuginone-resistant parasites. The rebuttal focuses on NCP26 which is not a triple-site inhibitor, but indeed these analogues are acting through PfcProRS as the principle functional target in vitro, and are ATP-competitive, proline-uncompetitive PfcProRS inhibitors, without much resistance generation.

We appreciate the reviewer's feedback. However, we cannot agree with this statement as the activities for all of the triple-site inhibitors reported in this manuscript against halofuginone-resistant parasites (halofuginone-induced and HFGR-I) are included in both our original submission and resubmission. We refer reviewer #4 to Figure 5j for the plots and to Table 1 and Supplementary Table 2 for the tabulated EC₅₀ values. This data was also discussed in the text at line 382 "Unfortunately, but not unexpected, MAT436, 35, 36, and 40 exhibited reduced

activity towards halofuginone-induced and HFGR-I-mutant parasites, comparable to halofuginone (Fig. 5j)."

(*Reviewer #4 – Comment #2*) In contrast, for MAT334 and 345 although not being triple-site inhibitors the authors already argued (in answering remark 7) that they demonstrated that the activity of MAT334 and MAT345 is reduced in halofuginone-resistant parasite strains (halofuginone-induced and HFGR-I), which have increased intracellular proline levels. The 20- to-40-fold decreased activity directly correlates with the ~20-fold-increase in proline in these parasite lines (Ref. 27).

It would have been nice therefore to see the effect of increased proline concentrations on the activity of real triple site inhibitors. Now the paper only states (further down around line 464) that the triple-site compounds bound both ProRS paralogs with subnanomolar affinity in the absence of either substrate (proline or ATP). Their high biochemical affinity translated well into antiparasitic activity against both ABS and liver stage Plasmodium parasites. The latter conditions obviously do contain competing proline.

This very point is addressed by our cellular data testing these inhibitors in halofuginone-resistant parasites (Figure 5j, Table 1, and Supplementary Table 2) which have elevated proline (halofuginone-induced) or both elevated proline and the *PfcProRS*^{L482H} mutation (HFGR-I). As noted above, we also compared the activity in wildtype and halofuginone resistant parasites, see line 382: "Unfortunately, but not unexpected, MAT436, 35, 36, and 40 exhibited reduced activity towards halofuginone-induced and HFGR-I-mutant parasites, comparable to halofuginone (Fig. 5j)."

Furthermore, we have shown the effect of proline on the triple-site inhibitors in a biochemical context with our TR-FRET assay. In both our original manuscript and in our resubmission, we have included the TR-FRET assay data for all of our compounds, including the triple-site inhibitors, in the absence of substrates and in the presence of 100 μ M proline. The plots for this were included in Supplementary Figure 6 and Supplementary Figure 10 and the K_D values are included in Table 1 (selected compounds) and Supplementary Table 2 (all compounds).

(*Reviewer #4 – Comment #3*) This first referee remark is followed by "7 major comments of reviewer #1" which in my opinion each time are adequately answered by an extensive rebuttal.

We are grateful to hear that the reviewer agrees that we completely addressed all major comments raised by reviewer #1.

(*Reviewer #4– Comment #4*) Only for the second paragraph of the rebuttal on remark 7 which discusses borrelidin not being a triple-site inhibitor, I have mixed feelings. The same statement is found on a number of places in the paper (in the abstract and within results as well as discussion) where the same is mentioned. The authors claim that borrelidin is a "tri-competitive" inhibitor that competes with all three substrates, but does not occupy the ATP-binding site. I

however immediately recalled a Nature Communications paper of 2015 (DOI: 10.1038/ncomms7402) on borrelidin which claims the compound to bind all three substrate binding sites [“a single molecule of borrelidin simultaneously occupies four distinct subsites within the catalytic domain of bacterial and human ThrRSs. These include the three substrate-binding sites for amino acid, ATP and tRNA associated with aminoacylation, and a fourth ‘orthogonal’ subsite created as a consequence of binding”]. The authors in their rebuttal refer to the same paper (PMID: 25824639). The same paper is referred to in the manuscript as well but only at the end (ref 49). We have to admit that the tRNA third site is only partially occupied by borrelidin in comparison with the new inhibitors of this manuscript, in which halofuginone is included in the new triple-site binding structures and therefore truly fully occupying all three substrate sites. I nevertheless am inclined to suggest the authors should slightly rephrase the various parts in the paper where the uniqueness of their triple-site inhibitors is mentioned. In my opinion, they should refer to the partial occupation of the third site by borrelidin.

We thank the reviewer for these comments and appreciate the opportunity to provide our rationale and justification for the wording of our claims. We would like to clarify that we are not contending that borrelidin does not occupy the tRNA-binding site of ThrRS, but based upon the rest of the reviewer’s comment, we assume that they are actually referring to the ATP-binding site.

Fang, Franklyn, Guo and coworkers report in their 2015 *Nature Communications* manuscript (Fang *et al.*) that borrelidin is a tri-substrate competitive inhibitor. As illustrated in Figure 2 of the Fang *et al.* paper, borrelidin barely extends to entrance of the adenosine binding pocket. In addition to Figure 2, we also provide a screenshot from Supplementary Movie 1 that was supplied with the Fang *et al.* paper, which shows the binding modes of borrelidin and ATP in the ThrRS active site.

In fact, Fang and Guo, in an excellent review (ref. 39 in our manuscript) that the authors published on the heels of the *Nature Communication* manuscript, discuss their own work on borrelidin including the borrelidin bound ThrRS structure in detail. Importantly, the authors explicitly state: “No triple active site (ATP-amino

acid-tRNA) inhibitor has been reported.” and that such ligands remain elusive. The authors go on to include a cartoon of what such a hypothetical compound might look like (no relationship to borrelidin) and elaborate upon why they hypothesized that then-hypothetical triple-site inhibitors would be appealing, but they do not reduce it to practice.

While we recognize the (intentionally?) ambiguous wording the authors chose in the 2015 *Nature Communications* manuscript, we believe that the reviewer, given this context, will agree that is appropriate to not second guess the authors' own words that establish unambiguous clarification.

The ATP-competitive inhibition mode of borrelidin can well be explained by an allosteric mode of inhibition, similar to the proline-competitive binding mode of MAT379, MAT334, and MAT345, which also do not bind the proline-pocket of ProRS. We provide detailed rationale for this mode of inhibition in our manuscript and the detailed rebuttal provided in response to the original reviews.

(Reviewer #4 – Comment #5) In addition, all minor comments (1-8) have been adequately dealt with and I specifically support inclusion of the very instructive figure 5 for the revised version (minor comment 2).

We are grateful for the reviewer's feedback and are glad that they appreciate the updated Figure 5.

Reviewer #2 comments:

“The experiment rationale and design are clear although the general ideas have been published earlier.”

(Reviewer #4 – Comment #6) The authors disagree with this simple but a bit negative statement of the referee, and I support the authors in that this manuscript really offers several new findings, and attempts to formulate solutions for the existing challenges. The authors further adequately answer both objections formulated by this referee in “result 4”. (reagent availability and limited impact of the work).

We are glad that the reviewer agrees that our response adequately addressed the previously raised concerns.

(Reviewer #4 – Comment #7) Indeed, development of selective triple site inhibitors of ProRS has not been achieved yet, but high in vitro selectivity for Plasmodium parasites has been shown for MAT436. As pointed out above it would have been desirable to see analogous tests also with the real triple site inhibitors.

We are not certain what this reviewer is asking here because such experiments have been reported in our manuscript and we provide a mechanistic explanation explain for the selectivity for *PfcProRS* over *HsProRS*, which can be rationalized

based upon the differential affinities of ATP and proline for these enzymes. To support this hypothesis, we reference representative examples for kinase inhibitors (ref 45 and ref 46 in our manuscript). Please also refer to the discussion provided in our previous response.

As noted above in response to reviewer 4's Comment #2, we included the TR-FRET assay data for all of our compounds, including the triple-site inhibitors, in the absence of substrates and in the presence of 100 μ M proline (see Supplementary Figure 6, Supplementary Figure 10, Table 1, and Supplementary Table 2). We also included cellular data for our triple site inhibitors against wildtype (Dd2-2D4) and halofuginone-resistant (halofuginone-induced and HFGR-I) ABS *P. falciparum* parasites, liver stage *P. berghei* parasites, and human hepatocytes (HuH7) in Figure 5i-k, Table 1, and Supplementary Table 2.

(Reviewer #4– Comment #8) Further, all 3 major issues raised by referee #2 have been answered at length, and likewise all smaller remarks have been adequately dealt with.

We thank the reviewer for their time and feedback.

(Reviewer #4) Further personal comments:

(Reviewer #4 – Comment #9) * Line 120 in the revised manuscript states: “Although no synthetic protocols for compound 2 or analogs have been reported”
However, synthesis of the Adachi compound (compound 2) was reported by L. Pang et al. in 2021 (<https://doi.org/10.3390/ijms22157793>), a reference paper which is quoted by the authors themselves for their structural arguments in their rebuttal.

The reviewer is correct that an alternative synthetic protocol for the synthesis of Compound 2 has been published a couple of months prior to the submission of our manuscript to *Nature Communications*. While we have cited this paper in our revised manuscript, we failed to notice that this information was provided in the Supplementary Information of the Pang et al. paper.

However, we would like to note that the synthesis of all compounds reported in our manuscript had been completed by the time the other manuscript was published. We furthermore would like to note that our synthetic procedures have been published already in 2019 within our PCT application (Ref. 47) and the structure of NCP26 was published and proline bound to *PfcProRS* (PDB 6T7K) was published in November 2020.

The sentence “Although no synthetic protocols for compound 2 or analogs have been reported” was intended to clarify that we did not fail to cite prior publications.

We therefore believe that the sentence was appropriate as is. However, we have changed it to:

“Compound **2** and analogs were readily accessible via a concise synthetic strategy starting from 3-aminopyrazine-2-carboxylic acid (see Supplementary Information).”

If this is not acceptable, we propose following alternative:

“Although no synthetic protocols for compound **2** or analogs had been reported at the time, we were readily able to access the target compound via a concise synthetic strategy starting from 3-aminopyrazine-2-carboxylic acid (see Supplementary Information). More recently another group reported a different synthetic approach.” **This would be followed by the citation to the L. Pang *et al.* paper.**

(Reviewer #4 – Comment #10) * Figure 5: MAT436 (33) should read compound #34 instead (typo)

We thank the reviewer for noting this oversight and will correct this mistake in our resubmission.

REVIEWERS' COMMENTS

Reviewer #4 (Remarks to the Author):

I thank the authors for their second revision and in particular for their (second) instructive rebuttal, which carefully addressed all points of all reviewers including my own small remarks (reviewer #4).

Indeed, the authors clearly pointed out that borrelidin is not a true triple site inhibitor in contrast to what I remembered personally based on the original publication of Fang and Guo in Nature Communications 2015.

Likewise, the authors revised their formulation in this latest revision regarding the synthetic protocols for the Adachi compound 2 and its various analogs.

As already stated previously, overall this is a very well written manuscript offering some potent new inhibitors against malaria parasites with in addition, the new FRET assay which was developed and which highlights a useful strategy for development of inhibitors for many other targets.

REVIEWER COMMENTS

Reviewer #4 (Remarks to the Author):

I thank the authors for their second revision and in particular for their (second) instructive rebuttal, which carefully addressed all points of all reviewers including my own small remarks (reviewer #4).

Indeed, the authors clearly pointed out that borrelidin is not a true triple site inhibitor in contrast to what I remembered personally based on the original publication of Fang and Guo in Nature Communications 2015.

Likewise, the authors revised their formulation in this latest revision regarding the synthetic protocols for the Adachi compound 2 and its various analogs.

As already stated previously, overall this is a very well written manuscript offering some potent new inhibitors against malaria parasites with in addition, the new FRET assay which was developed and which highlights a useful strategy for development of inhibitors for many other targets.

We gain would like to express our gratitude to the reviewer for their time, through review and constructive criticism. We appreciate the positive feedback. We are glad that we were able to fully address all remaining questions.